# Individual differences in avoiding feelings of disgust: Development and construct validity of the disgust avoidance questionnaire

Paula von Spreckelsen[1]*, Nienke C. Jonker[1], Jorien Vugteveen[2], Ineke Wessel[1], Klaske A. Glashouwer[1,3], Peter J. de Jong[1]

1 Department of Psychology (Expertise Group: Clinical Psychology and Experimental Psychopathology), University of Groningen, Groningen, The Netherlands, 2 Department of Psychology (Expertise Group: Psychometrics and Statistics), University of Groningen, Groningen, The Netherlands, 3 Department of Eating Disorders, Accare Child and Adolescent Psychiatry, Groningen, The Netherlands

* p.von.spreckelsen@student.rug.nl

**Data Availability Statement:** All data files (Study 1, Study 2) are available on the OSF (URL: https://osf. io/qnfxg/ ; DOI: 10.17605/OSF.IO/QNFXG).

## Abstract

We developed and examined the construct validity of the *Disgust Avoidance Questionnaire* (DAQ) as a measure of people's inclination to prevent experiencing disgust (disgust prevention) and to escape from the experience of disgust (disgust escape). In a stepwise item-reduction (Study 1; $N = 417$) using Exploratory Factor Analysis (EFA) based on a 4-subscale distinction (behavioral prevention, cognitive prevention, behavioral escape, cognitive escape), we selected 17 items from a pool of potential items. In order to incorporate the conceptual overlap between dimensions of disgust avoidance, focus (prevention vs. escape), and strategy (behavioral avoidance vs. cognitive avoidance), we specified an adapted model. In this model, we allowed each item to load on one type of dimension and one type of strategy, resulting in four overlapping factors (prevention, escape, behavioral avoidance, cognitive avoidance). Evaluation of this overlapping 4-factor model (Study 2; $N = 513$) using Exploratory Structural Equation Modeling (ESEM) and Confirmatory Factor Analysis (CFA) showed promising model fit indices, factor loadings, factor correlations, and reliability estimates for three of the four factors (prevention, behavioral avoidance, cognitive avoidance). Those three subscales also showed good convergent validity. In contrast, the results related to the escape factor may call the suitability of self-report to assess disgust escape into question. In light of the exploratory nature of the project, future examinations of the DAQ's validity and applicability to more diverse samples are essential. A critical next step for future research would be to examine the DAQ's criterion validity and the distinctive roles of the DAQ subscales in (clinical) psychological constructs and processes.

## Introduction

Besides the subjective feeling of aversion (affective component), the experience of disgust involves cognitive processes (e.g., appraising/interpreting an object as disgusting), a physiological reaction (e.g., feelings of nausea and a characteristic facial expression), and an urge to

**Funding:** Preparation of this article by Klaske Glashouwer was supported by a Veni grant [451-15-026] awarded by the Netherlands Organization for Scientific Research (NWO). The funder has played no role in the research.

**Competing interests:** The authors have declared that no competing interests exist.

literally or figuratively distance the self from the repulsive object [e.g., 1]. Disgust plays a vital role in our health and survival by protecting us from pathogens, thereby acting as a 'disease avoidance system' [e.g., 2,3]. Accordingly, the experience of disgust is most strongly tied to stimuli that are somehow associated with an increased risk of the transmission of infectious diseases, including rotten food, bodily products, insects, or dead bodies (pathogen disgust; [4]). Next to pathogens, disgust is also commonly elicited by certain sexual stimuli (e.g., sexual acts/partners) that signal threats to a person's reproductive success (sexual disgust; [4]) as well as by social transgressions (e.g., lying, cheating, interpersonal violence; moral disgust; [4]). While some disgust-elicitors appear to be relatively universal, personal experiences, social standards, and cultural beliefs can have a strong influence on what we appraise as repulsive [5].

Although disgust generally plays a vital role in promoting our health and survival, under certain circumstances disgust may become maladaptive and impede normal functioning. In line with this notion, research indicates that disgust is involved in several forms of psychopathology, including anxiety disorders (e.g., blood-injection-injury phobia, spider phobia), obsessive compulsive disorder (OCD), post-traumatic stress disorder, eating disorders, sexual dysfunctions, schizophrenia, and hypochondriasis [see 6]. On a broader societal level, disgust also seems to play a role in stigma, extreme prejudice/bigotry, de-humanization, and discrimination [7–13]. However, the emotion of disgust has received less attention than other emotions in research so far [14]. Due to the implications of disgust on both the individual and societal level, more research is needed to understand the human disgust experience.

Research has so far identified two main dimensions of the disgust experience. One of the dimensions, termed *'disgust propensity'*, refers to a person's tendency to easily experience disgust, the other dimension termed '*disgust sensitivity'*, describes a person's tendency to appraise the experiencing of disgust as aversive [15]. Both of these individual difference variables, if experienced in excess, seem to be associated with symptoms of mental disorders [6]. The relevance of disgust propensity and sensitivity to psychopathology has been proposed to be related to motivating people to avoid disgust-eliciting stimuli, thus increasing fear and avoidance of phobic stimuli [16]. Research found that disgust propensity and/or sensitivity are indeed predictive of behavioral and visual avoidance of disgusting stimuli [16–21]. It has been suggested that such avoidance responses not only may help prevent exposure to (potential) pathogens, but also help regulate the emotional experience of disgust [cf. 22].

## Disgust avoidance

We propose that individual differences in *disgust avoidance* may be an important, yet unexplored, third dimension that is relevant to the study of the human disgust experience. We define disgust avoidance as a person's tendency or inclination to avoid experiencing disgust. Just like disgust propensity and disgust sensitivity, disgust avoidance is expected to differ across individuals. In general, people tend to appraise the experience of disgust as aversive and try to avoid it. However, disgust can carry a peculiar attraction or amusement [23], and a disgusting stimulus may also appear fascinating [24]. For example, a TV program showing a surgery openly displaying the insides of the human body, including organs, veins, blood, and other fluids may intrigue, thrill, or nauseate its viewers. Research indicates that the experience of disgust can be accompanied by enjoyment [25] or humor [26,27]. The experience of disgust can have an appetitive quality to some people and be experienced as highly aversive by others (i.e., people with a high disgust sensitivity; cf. [28]).

Individual differences in disgust avoidance are likely to be closely linked to individual differences in disgust propensity and sensitivity. In other words, a person who is easily repulsed and appraises experiencing disgust as highly aversive will likely show a heightened tendency to

avoid experiencing it. Despite the close relation, disgust avoidance represents a unique construct because it encompasses the inclination to approach vs. avoid aversive emotional states, specifically the emotion of disgust. Correlation coefficients found between disgust propensity/sensitivity and behavioral/visual avoidance of disgusting stimuli seem to fall in a range of around .25 to .70 [16–21]. It therefore appears that avoidance of disgusting stimuli cannot completely be accounted for by individual differences in disgust propensity and sensitivity. Similarly, we would not expect individual differences in disgust avoidance to be accounted for by disgust propensity/sensitivity. By representing people's tendencies to avoid (vs. approach) the emotional state of disgust, we believe that disgust avoidance can provide insights into (dysfunctional) psychological processes beyond what can be learned from examining disgust propensity and sensitivity.

Disgust associations appear to be highly persistent and resistant to extinction [e.g., 29–34]. By preventing exposure to disgust-eliciting stimuli, disgust avoidance obstructs the learning of new associations, thus leading to the persistence of the disgust association. In general, disgust avoidance may be considered as an adaptive response that serves to distance oneself from stimuli signaling contamination threats. However, when disgust is experienced in excess–especially in response to a disgust-elicitor that does not represent a real threat (e.g., own body fat)–the tendency to avoid experiencing disgust can be maladaptive. For example, patients with an eating disorder, may attempt to avoid experiencing disgust in response to their own body fat by restricting their intake of (high-caloric) food items, excessive dieting, and engaging in purging behavior [cf. 35]. Although the specific stimulus may differ, such a process could also apply to other disgust-relevant disorders. For example, avoidance behavior in individuals with a specific phobia or sexual dysfunction may represent attempts to avoid exposure to intense disgust experiences elicited through the phobic stimulus such as a spider [e.g., 36] or sexual intercourse [e.g., 37], respectively. As a result of this avoidance, we would expect maladaptive disgust associations to become more persistent, contributing to the maintenance of psychopathology.

Avoidance of internal experiences (e.g., of emotions, cognitions; 'experiential avoidance') is common in many mental disorders and seems to be an important factor maintaining and exacerbating symptomatology [e.g., 38]. Disgust avoidance can be conceptualized as a specific form of experiential avoidance (i.e., specifically relating to the emotion of disgust) that may be especially relevant in the development and persistence of disgust-related disorders such as OCD. For example, one study found that disgust avoidance in the context of contamination fear was associated with a number of OCD symptoms [22]. This study also found that the motivation to avoid disgust was associated with other OCD symptoms than the motivation to avoid harm. In addition, traditional treatment approaches (e.g., exposure) seem to be less efficient in the context of disgust-based OCD symptoms [32,33]. This highlights the differential role of emotions in psychopathology, and that specifically focusing on the avoidance of disgust may help understand individual differences in symptoms of psychopathology. Next to the importance of distinguishing between disgust and other emotions, it seems also important to distinguish between different motivational foci within the concept of disgust avoidance. More specifically, we hypothesize that disgust avoidance operates both at a reflective (prevention) and a reactive (escape) level.

**Prevention-focused disgust avoidance.** The urge to avoid exposure to a disgusting cue has been described as an integral part of the disgust response and may thus be triggered rather automatically [e.g., 1]. However, when the goal shifts from avoidance of external stimuli that signal contamination threats to the avoidance of experiencing the feeling of disgust, more strategic processes may come into play. As such, disgust avoidance may be seen as a form of emotion regulation strategy. According to Gross [39], antecedent-focused coping refers to emotion

regulation strategies that occur before an emotion is experienced. One may refer to antecedent-focused coping as engagement in behaviors or cognitions that aim to prevent a negative emotion from being experienced. Translating this to the domain of disgust, disgust avoidance that is prevention-focused ('*disgust prevention*') may be seen as a strategic form of cognitive or behavioral avoidance that aims at preventing the experience of disgust altogether.

Although the inclination to prevent experiencing disgust may generally be seen as adaptive, an excess in disgust prevention is expected to be detrimental. The perspective that pathogen disgust evolved to protect humans from pathogens that cannot be seen or otherwise detected may partially explain why disgust is geared towards a *better safe than sorry* heuristic. In case of life or death, it seems wise to play it safe. Such an adaptive conservatism may give rise to a high false alarm rate and may promote the generalization of disgust to non-threatening stimuli [2,3]. In addition, disgust appears to operate according to the laws of sympathetic magic [40]. This means that disgust can easily transfer from a disgust-elicitor to a neutral object (law of contagion), and that disgust can be triggered by an object resembling a disgust-elicitor (law of similarity). Because of these qualities, any given situation may carry the danger of experiencing disgust. Thus, people with a strong tendency to prevent experiencing disgust are likely to engage in extreme avoidance of various situations or might resort to unhealthy avoidance strategies (e.g., extreme dieting to prevent experiencing disgust to own body fat). In sum, relatively strong disgust prevention could play a crucial role in problematic avoidance patterns, isolation, and unwillingness to seek treatment, thus exacerbating psychopathology.

**Escape-focused disgust avoidance.**   Disgust avoidance may also take a more reactive form. Next to antecedent-focused emotion regulation strategies, emotion regulation theorists have described response-focused coping. This type of coping refers to strategies people use to deal with emotions once they are elicited [39]. In the case of negative emotional states, response-focused coping can refer to strategies aiming at escaping from such undesirable emotional states. According to theoretical viewpoints on disgust, experiencing disgust instinctively results in the expulsion of or distancing from the disgusting stimulus [e.g., 2]. Therefore, as a reactive form of coping, we expect that an escape-focused disgust avoidance ('*disgust escape*') represents a rather automatic form of disgust avoidance. Yet, people may still vary in the strength of their reflexive inclination to escape from stimuli that elicit disgust.

Disgust escape is adaptive when it promotes people to distance themselves from situations in which a threat to the organism is imminent. However, having a strong tendency to quickly escape from disgust would impede people to identify a false alarm and maintain the disgust-eliciting quality of a given stimulus. Moreover, by aborting the experience of disgust as quickly as possible, people would not be able to gain a sense of control over their disgust experience, making it seem even more overwhelming and intolerable. In the context of psychopathology, a heightened disgust escape is thus assumed to contribute to the maintenance and exacerbation of disgust associations that play a role in several disorders. Lastly, disgust escape may be a factor impeding the success of exposure treatment, which is a common treatment strategy for a number of disgust-related disorders (e.g., anxiety disorders [41]; eating disorders [42]).

## The current project

In sum, we propose that high trait disgust avoidance may play an important role in the development and persistence of disgust-relevant psychopathology. Measuring people's tendency to avoid experiencing disgust may help us to draw a more refined picture of experiential avoidance processes in disgust-related disorders. Such a measure can also help clarify the role of disgust sensitivity in these psychopathologies (i.e., disgust avoidance representing the mechanism through which disgust sensitivity relates to psychological suffering). Distinguishing between

disgust prevention and disgust escape may further help us understand individual differences between and within different forms of mental disorders. For example, the two forms of disgust avoidance may be related to different symptomatic avoidance behaviors (that can be characteristic of different diagnostic categories within one group of disorders). Although we would expect the two dimensions to be highly correlated, they might show different developmental trajectories within a given mental disorder (e.g., initial elevated disgust escape results in increased disgust prevention over time). Lastly, the distinction between disgust prevention and escape may also be of relevance to the treatment of disgust-related disorders, making it possible to identify which avoidance strategy to focus on during, for example, exposure interventions.

Existing measures so far do not explicitly index the strength of people's habitual inclination to avoid disgust (i.e., trait disgust avoidance). Thus far, most research has utilized behavioral avoidance tasks (BATs) to measure the extent to which people avoid disgusting stimuli [e.g., 16–21]. This assessment method, however, does not specifically measure the extent to which people avoid the emotional experience of disgust (vs. avoid the disgusting stimulus). Furthermore, BATs primarily assess behavioral avoidance at the state level and are dependent on the specific stimulus used in the task. Disgust avoidance has also been assessed in the form of a questionnaire on contamination fear in OCD (Contamination Fear Core Dimension Scale; CFCDS; [21]). In addition to being restricted to contamination concerns, this scale combines the motivation to avoid disgust and the tendency to fear disgust, and therefore does not represent a pure measure of disgust avoidance.

We therefore designed the *Disgust Avoidance Questionnaire* (DAQ), which aims to measure the strength of people's inclination to prevent experiencing disgust (disgust prevention) and their inclination to escape from experiencing disgust (disgust escape). Building on existing stimulus-independent measures of trait disgust (e.g., the Disgust Propensity and Sensitivity Scale; DPSS), the DAQ assesses individual differences in disgust avoidance independent of specific disgust elicitors. In this article, we report on the development and the psychometric properties of the DAQ in two samples of young adults. First, we selected a number of potential items (from existing measures in the field of experiential avoidance and disgust) to be condensed with a stepwise item-reduction method (Study 1). Subsequently, we examined the factor structure and the practical applicability of the reduced item set (Study 2). We also examined the DAQ's convergent validity by examining the extent to which the DAQ is associated with other disgust-related and emotion-regulation measures (Study 2).

## Study 1: Item selection of the DAQ

The main goal of Study 1 was to select items for the DAQ using both 'judgmental'/evaluative (e.g., item content, wording, etc.) and statistical criteria (e.g., item loadings, reliability estimates; cf. [43]). We first compiled a list of potential items for the DAQ (based on judgmental criteria) and subsequently condensed it through a stepwise item reduction. The goal of the step-wise item reduction was to find a coherent item set per hypothesized subscale of the DAQ and it was based mainly on statistical criteria. More specifically, we used single- and multi-factor EFA (*exploratory factor analysis*; [44]) models and fitted them on the items of each hypothesized subscale to exclude 'suboptimal' items with the goal to create unidimensional factor models per subscale. As a last step, we fitted an EFA on all items to examine whether the item loadings were in line with our hypothesized subscales. We aimed for a sample size of at least 400 participants, based on Fabrigar and colleagues [45] categorizing sample sizes of $N > 400$ as large.

## Method

**Participants.** We recruited our sample via two university-based participant pools consisting of (Pool 1) first-year bachelor psychology students ($n = 162$; participation in exchange for course credit) and (Pool 2) a broader group of young adults ($n = 255$; participation in exchange for financial compensation: 2€). The participants (total $N = 417$; 77.2% female) were tested between November 2017 and January 2018. The majority of participants were in their early twenties, either Dutch or German, and studied Psychology (see Table 1 for sample characteristics). From the initial $n = 495$, $n = 78$ (15.76%) participants were excluded, because they (a) did not consent to participate in the study/wanted to withdraw their responses from the study ($n = 20$; 25.64%), did not answer both control questions correctly ($n = 58$; 74.36%).

**Materials.** The initial item set of the *Disgust Avoidance Questionnaire* (DAQ; initial item set) consisted of 25 items assessing people's tendency to avoid experiencing disgust. We aimed to base the wording of DAQ items on items used in the field. Items from the Multidimensional Experiential Avoidance Questionnaire (MEAQ; [46]), the Emotional Avoidance Questionnaire [EAQ; 47], and the Cognitive Avoidance Questionnaire (CAQ; [48]) were taken as a representative sample of items that are commonly used in avoidance questionnaires. The type and wording of these items were used as a framework to generate an item pool for the DAQ. For this item pool, we only selected items that were either prevention-focused (i.e., referring to an action aiming at preventing adversity) or escape-focused (i.e., referring to an action aiming to escape from adversity). We also made sure that the pool included both behavioral avoidance (physically avoiding an activity, situation, object, or place) and cognitive avoidance (suppression of or distraction from negative emotions/thoughts) in order to represent both strategies through which a person can engage in disgust avoidance.

The items were then adapted such that they referred to the avoidance of a content-independent cue (i.e., situation, activity, thought) that can elicit feelings of repulsion. As an example of

**Table 1. Gender, age, nationalities, and study fields of Study 1 (overall and per recruitment pool).**

|  | Overall ($N = 417$) | Pool 1 ($n = 162$) | Pool 2 ($n = 255$) |
|---|---|---|---|
| Age (Mean, SD)[1] | 21.80 (4.39) | 20.25 (3.12) | 22.80 (4.78) |
| Gender[1] |  |  |  |
| Female | 77.2% | 73.5% | 79.6% |
| Male | 21.8% | 25.3% | 19.6% |
| Genderqueer | 0.2% | 0.6% | 0% |
| Nationality |  |  |  |
| Dutch | 43.6% | 37.0% | 47.8% |
| German | 29.0% | 40.7% | 21.6% |
| Other[2] | 27.4% | 22.3% | 30.6% |
| Field of Study |  |  |  |
| Psychology | 61.2% | 99.4% | 36.9% |
| Other[3] | 32.6% | 0.6% | 52.9% |
| Not Studying | 6.2% | 0% | 10.2% |

*Note.*

Pool 1 = participation in exchange for course credit.

Pool 2 = participation in exchange for financial compensation.

[1]Responses were missing for $n = 3$.

[2]Other included a variety of nationalities (e.g., English, Eastern & Southern European, Asian, Baltic, Scandinavian).

[3]Other included a variety of study fields (e.g., Biology, Medicine, Communications, Law, Finance, Economics).

how an original item was adapted to express repulsion, '*I won't do something if I think it will make me feel uncomfortable*' (MEAQ) was changed to '*I won't do something If I know it will be revolting*' (DAQ). Further, the wording of some items was adapted to make their focus on prevention or escape clearer. The resulting item set consisted of 25 items aiming to measure people's tendency to avoid experiencing disgust through prevention (13 items; e.g., '*I won't do something if I know it will be revolting*') or escape (12 items; e.g., '*I am quick to stop any activity that makes me feel disgusted*'). All source items and adapted DAQ items can be found in S1 Table (accessible on the OSF: https://osf.io/qnfxg/). Prevention- and escape-focused items were presented in alternating order (i.e., p-e-p-e-p-e-. . .) to avoid artificially inflating item error correlations. A short instruction was included, which illustrated examples of general disgust elicitors (pathogen, sexual, and moral disgust cues). A 7-point Likert scale ranging from '*strongly disagree*' (1) to '*strongly agree*' (7) was chosen as the response system. The item set and instructions can be found in S1 Appendix (in the order of presentation; accessible on the OSF: https://osf.io/qnfxg/).

The materials also included the initial item set (25) of the *Body-related Disgust Avoidance Questionnaire* (B-DAQ) which aims to assess people's tendencies to avoid experiencing body-related disgust. The B-DAQ is a body-related version of the Disgust Avoidance Questionnaire (DAQ). The B-DAQ and related materials can be found on the OSF (https://osf.io/4mzfs/) and will not be described here because it would extend the scope of the paper.

**Procedure.**   After receiving approval from the Ethics Committee of the University of Groningen (Approval code: 17117-SP-NE), advertisements for the study were posted on online platforms (Facebook, university-based participant pools), which included a short description of the study and a link that forwarded the participants to the online questionnaires in Qualtrics (Qualtrics, Provo, UT). In Qualtrics, participants were informed about the study (its general aim and content) and asked to give consent to participate in the study. Participants filled out the DAQ. Subsequently they completed the B-DAQ (available on the OSF: https://osf.io/4mzfs/). Two control questions were included (one in in the DAQ and one in the B-DAQ), which asked participants to select a specific answer category (e.g., please click the left-most answer option) that served as a check to exclude inattentive participants. Lastly, participants were given the possibility to leave notes concerning the questions they just answered, filled in demographic details (i.e., age, gender, field of study), and were given the option to indicate whether they would like to withdraw their responses from the study. The Qualtrics session ended with debriefing information about the goal of the study. Participation lasted around 15 minutes.

**Analysis.**   Informed by theory, we set out to distinguish two separate but related subscales: Prevention and escape. A stepwise item reduction method was used per intended factor by means of *ordinal* EFA with *Oblimin rotation* in Mplus version 8.0 [49]. We performed an ordinal factor analysis because of a high likelihood that item distributions were not Normal (Likert-scale response format). An Oblimin rotation method was chosen because the factors were expected to be correlated (*r* estimated between .60 - .80). The goal was to create unidimensional factor models per subscale, through the stepwise exclusion of 'suboptimal' items. Initially, single- and multi-factor EFA models were fitted on the item set of each of the two presumed factors.

Based on our presumed 2-factor structure, we separated the initial set of prevention-focused items (13) from the initial set of escape-focused-items (12; see Fig 1A) and conducted the stepwise item reduction, beginning with estimating single- and multi-factor EFA models, per subscale. For the prevention factor, 9 out of the initial 13 items loaded on one factor and the remaining items loaded on another factor (see S4 for model fit statistics and factor loadings of a 2-factor EFA model; accessible on the OSF: https://osf.io/qnfxg/). On closer inspection of the

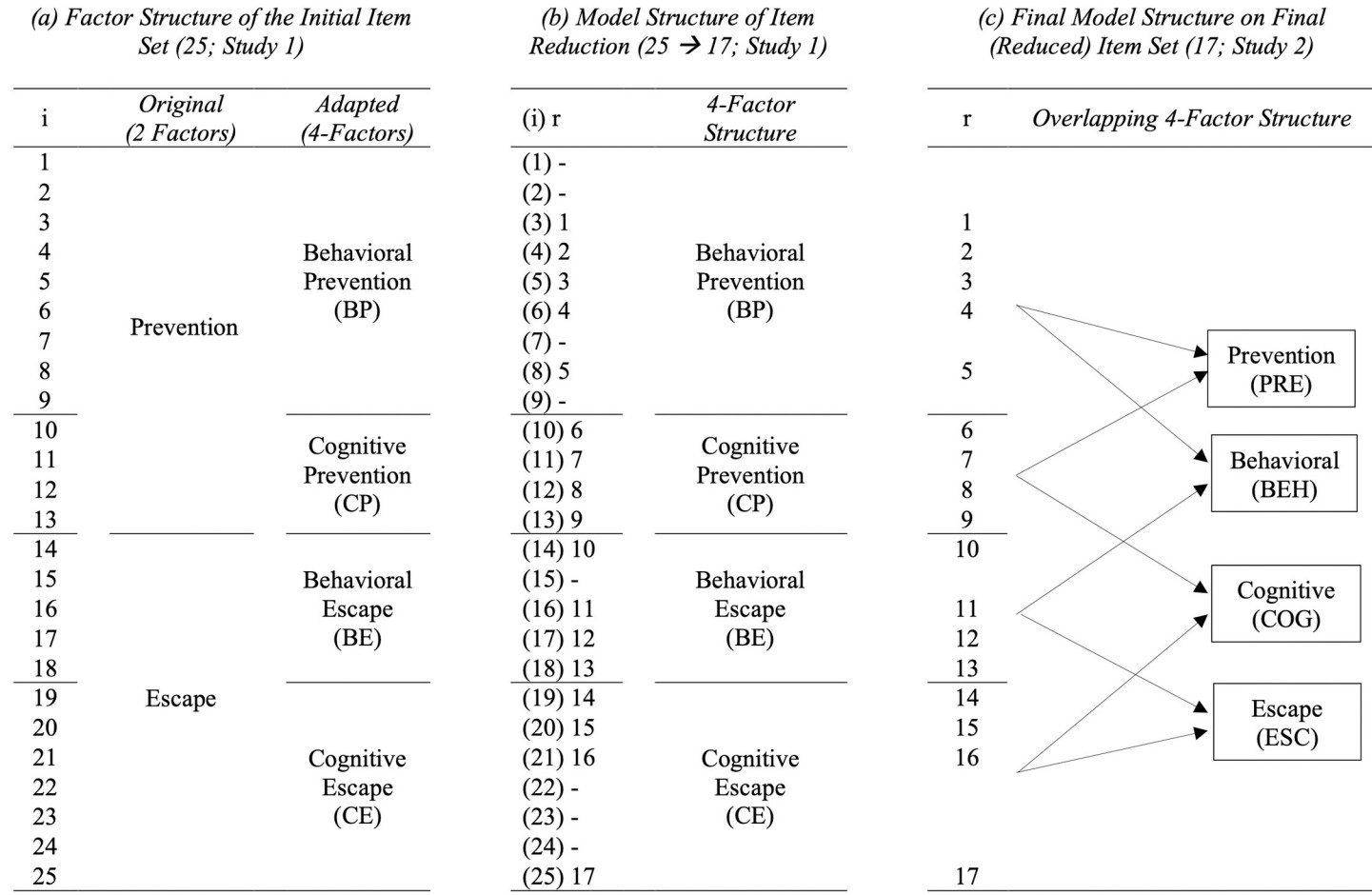

**Fig 1.** Changes in DAQ factor structure and corresponding item numbers in Study 1 (a, b) and Study 2 (c). i: Initial item number. r: Reduced item number.

items, we noticed that the 9 items referred to *behavioral* avoidance (e.g., "I try to avoid activities that could make me feel disgusted") and the remaining items referred to *cognitive* avoidance (e.g., "I try hard to avoid thinking about a past repulsive situation"). Similar results were found for the escape subscale: five out of the initial 12 items (e.g., "If I start feeling strong disgust, I prefer to leave the situation") loaded on one factor that regarded *behavioral* avoidance while remaining items (e.g., "If thoughts about disgusting things cross my mind, I try to push them away as much as possible") loaded on another factor regarding *cognitive* avoidance (see S3 Table for model statistics and item loadings of a 2-factor EFA model; accessible on the OSF: https://osf.io/qnfxg/). It thus seemed that the items aimed at measuring prevention and escape can be distinguished on a behavioral and cognitive level. We therefore decided to split each intended factor (i.e., prevention and escape) into two, resulting in four factors: (1) Behavioral prevention (items 1–9), (2) Cognitive prevention (items 10–13), (3) Behavioral escape (items 14–18), and (4) Cognitive escape (items 19–25). The initially presumed two-factor structure and the adapted 4-factor structure can be found in Fig 1A. Please note that item numbers given here refer to the initial item set as presented in Fig 1A.

Based on the adapted four factor structure, we fitted single- and multi-factor EFA models on the item set of each of the four presumed factors. In case of suboptimal model fit of the one-factor model (*comparative fit index* [CFI; 50] < .90, and *root mean square error of*

*approximation* [RMSEA; 51] >.08), multi-factor EFA models were examined to identify an item to be excluded from the presumed subscale. More specifically, the item with the lowest item target loading (i.e., low loading on the 'intended'/same factor as the other items; leading criterion) and/or highest cross-loading (i.e., loading on an unintended/separate factor) was identified and excluded from the item set. This procedure was repeated until the one-factor EFA solution showed acceptable fit indices (CFI ≥ 0.90, RMSEA ≤ 0.08), and all retained items had good target- and no cross-loadings. We considered target loadings of <0.3 insufficient, 0.3–0.4 acceptable, and >0.4 good, and cross-loadings of >0.4 problematic and >0.3 questionable. Cases in which the decision could not be made based on item loadings only (e.g., two items with similarly low target loadings) were also evaluated based on the representation of the conceptual theme of the factor, similarity to other items, distribution, wording, and conceptual overlap with other constructs.

For the reduced item sets, we were aiming to retain 4–5 items per subscale, a good fit of the one-factor EFA model (CFI ≥ 0.90, RMSEA ≤ 0.08), at least acceptable reliability (Ω ≥ 0.7; [52]) and a cohesive and sufficiently broad coverage of the concept of interest. The Omega Coefficient (Ω) was calculated to assess the reliability of the continuous variable underlying the observed categorical variables.

As the last step, the distinction between the reduced sets of the presumed subscales was assessed by fitting a 4-factor EFA model on the combined item sets. We examined the factor loadings (same item loading thresholds as described above) and fit (CFI, RMSEA) of this multi-factor EFA model, in which items belonging to each subscale should load on their intended factor only.

## Results

**Stepwise item reduction.** In total, the initial number of 25 items was reduced to a *Final Item Set* of 17 (i.e., 8 items were excluded), as can be seen in Fig 1B. Please note that item numbers referred in the description of the item reduction correspond to the initial 25-item set (see Table 2). For subscale 1 *Behavioral Prevention (BP)*, the total number of 9 items was reduced to 5 items (see Table 2). Reasons for excluding items 1, 9, 7, and 2 included: high loadings on a different separate factor, conceptual overlap with other items, distributional problems (use of a restricted range of answer options), and lowest target loading. The resulting item set of 5 items had high target loadings, excellent fit, and reliability statistics (see Table 2). All 4 items of subscale 2 *Cognitive Prevention (CP)* had high loadings on a one-factor EFA model (see Table 2). Although the CFI indicated a good fit, the RMSEA did not. Exclusion of item number 10 (the item with the lowest loading) would have resulted in better fit indices for a three-item solution. However, deleting one item would have prevented having a sufficient number of items for the subscale. The reliability was good for the full item set. With regard to the RMSEA, it has been suggested that in case of high reliability and small specific variance of the variables/items, RMSEA can reject the model when there is only minor model error [53]. Therefore, we decided to stick with the full item set (see Table 2) despite the insufficient RMSEA value.

The 5-item set of subscale 3 *Behavioral Escape (BE)* was reduced to a set of four items. Item number 15 was excluded because it had a low target loading and a high loading on a separate factor. The CFI indicated a good fit of the resulting 4-item set but the RMSEA did not. At the expense of reaching an RMSEA value of ≤ 0.08 by excluding the next item, we retained the 4-items set (see Table 2) to ensure a sufficient number of items. The reliability was good for the reduced item set. The initial 7-item set of subscale 4 *Cognitive Escape (CE)* was reduced to a set of 4 items. Items number 24, 23, and 22 were excluded due to low target loadings and high loadings on separate factors. The resulting 4-item set (see Table 2) had an excellent fit

**Table 2. Standardized factor loadings, fit indices, and reliabilities of the 1-factor EFA models per subscale for the reduced item sets of the DAQ ($N$ = 417).**

| Item Number | | Item Wording | Factor Loadings |
|---|---|---|---|
| i[a] | r[b] | | |
| *Behavioral Prevention (BP; CFI = 0.998, RMSEA = 0.050 [90% CI: 0.000–0.094]; Ω = .90)* | | | |
| 1 | - | [I rarely do something if there is a chance that it will disgust me.] | - |
| 2 | - | [I won't do something if I know it will be revolting.] | - |
| 3 | 1 | I try to avoid activities that could make me feel disgusted. | **.77** |
| 4 | 2 | I avoid actions that remind me of repulsive things. | **.84** |
| 5 | 3 | I try hard to avoid situations that might bring up feelings of repulsion in me. | **.84** |
| 6 | 4 | I avoid certain situations that make me pay attention to disgusting things. | **.76** |
| 7 | - | [I avoid situations if there is a chance that I will feel revolted.] | - |
| 8 | 5 | I avoid objects that can trigger feelings of disgust. | - |
| 9 | - | [I avoid places that make me think of things that disgust me.] | **.79** |
| *Cognitive Prevention (CP; CFI = 0.996; RMSEA = 0.122 [90% CI: 0.068–0.184]; Ω = .90)* | | | |
| 10 | 6 | I try not to think about gross situations. | **.73** |
| 11 | 7 | I try hard to avoid thinking about a repulsive past situation. | **.84** |
| 12 | 8 | I distract myself to avoid thinking about things that disgust me. | **.86** |
| 13 | 9 | To avoid thinking about things that revolt me, I force myself to think about something else. | **.87** |
| *Behavioral Escape (BE; CFI = 0.998; RMSEA = 0.095 [90% CI: 0.040–0.159]; Ω = .89)* | | | |
| 14 | 10 | I am quick to stop any activity that makes me feel disgusted. | **.79** |
| 15 | - | [If I am doing something that makes me feel repulsion, I prefer to stop that activity.] | - |
| 16 | 11 | If I start feeling strong disgust, I prefer to leave the situation. | **.71** |
| 17 | 12 | If I am in a situation in which I feel revolted, I leave the situation immediately. | **.82** |
| 18 | 13 | I am quick to leave any situation that makes me feel disgusted. | **.93** |
| *Cognitive Escape (CE; CFI = 1.000; RMSEA = 0.000 [90% CI: 0.000–0.065]; Ω = .92)* | | | |
| 19 | 14 | When I think about something gross, I push those thoughts out of my mind. | **.83** |
| 20 | 15 | When thoughts about repulsive things come up, I try very hard to stop thinking about them. | **.88** |
| 21 | 16 | If thoughts about disgusting things cross my mind, I try to push them away as much as possible. | **.95** |
| 22 | - | [If I feel disgusted or think about something repulsive, I try to distract myself.] | - |
| 23 | - | [I usually try to distract myself when I feel disgusted.] | - |
| 24 | - | [When memories of disgusting experiences come up, I try to focus on other things.] | - |
| 25 | 17 | When thoughts about revolting things come up, I try to fill my head with something else. | **.78** |

*Note.*

a: i = initial item set

b: r = reduced item set; Factor loadings of $\geq 0.3$ are marked bold. The one-factor EFAs show item loadings on a 1-factor EFA model evaluated per subscale (BP, CP, BE, CE).

and reliability. In some cases, extreme CFI/RSMEA values might indicate model identification problems, but inspection of the Chi-Square test of model fit indicated that this was not the case for the CE subscale ($X^2(2) = 0.53$, $p = .768$).

*Examination of combined item set after item reduction.* As the last step, the fit of a 4-factor EFA model on those selected items was evaluated. The results can be seen in Table 3. The 4-factor EFA solution had an acceptable fit (CFI = 0.990, RMSEA = 0.074 [90% CI: 0.064–0.084]). Most items (15) displayed acceptable target loadings of > .30. However, six items showed cross-loadings of > .30, of which two items exhibited low target loadings of < .30. We did not exclude any additional items at this stage because we did not set the factor loadings in the 4-factor EFA model as a criterion to exclude items.

**Table 3. Standardized factor loadings of a 4-factor EFA model on the combined item set after item reduction (*N* = 417).**

| Items | | Four-factor EFA | | | |
|---|---|---|---|---|---|
| | | **1** | **2** | **3** | **4** |
| 1 | Behavioral Prevention (BP) | .19 | -.12 | **.52** | **.34** |
| 2 | | **.43** | -.04 | **.33** | .24 |
| 3 | | **.81** | .03 | .08 | .06 |
| 4 | | **.30** | .01 | **.36** | .28 |
| 5 | | **.49** | .02 | .21 | .22 |
| 6 | Cognitive Prevention (CP) | .06 | **.42** | .07 | **.33** |
| 7 | | .29 | **.60** | -.04 | .12 |
| 8 | | .11 | **.79** | -.02 | -.00 |
| 9 | | -.02 | **.90** | -.00 | .05 |
| 10 | Behavioral Escape (BE) | .05 | .10 | **.74** | -.03 |
| 11 | | -.09 | -.07 | **.78** | .20 |
| 12 | | **.39** | .21 | **.50** | -.19 |
| 13 | | .15 | .15 | **.77** | -.14 |
| 14 | Cognitive Escape (CE) | -.05 | .15 | .12 | **.71** |
| 15 | | .18 | .25 | -.07 | **.63** |
| 16 | | .10 | .22 | .02 | **.72** |
| 17 | | -.12 | **.88** | .11 | .10 |

*Note.* Factor loadings of $\geq 0.3$ are marked bold. The four-factor EFA shows item factor loadings on a 4-factor model evaluated in the combined item set.

## Study 2: Factor structure of the DAQ and relationships with other constructs

### Factor structure

The overall fit of the 4-factor EFA described in Study 1 was acceptable, but there were some problematic target- and cross-loadings. We set out to evaluate whether the 4-factor EFA of Study 1 could be retained in a new sample using *confirmatory factor analysis* (CFA; [54]) performed in Mplus (version 8.0; [48]). In CFA, items are allowed to load on their intended factor (s) only: cross-loadings are fixed to zero, which is the typical approach to evaluate an instrument's internal structure. After running into several problems when examining the 4-factor EFA in a CFA framework (problems with the latent variable covariance matrix, possibly indicating model misspecification; sub-optimal fit indices), we reconsidered our statistical approach to modeling the internal structure of the DAQ.

**Overlapping 4-factor model.** There are four concepts that we hypothesized to be underlying our statistical model: *disgust prevention* (PREV), *disgust escape* (ESC), *behavioral disgust avoidance* (BEH), and *cognitive disgust avoidance* (COG). These four concepts (PREV-ESC--BEH-COG) could be argued to represent two dimensions of disgust avoidance, namely focus (PREV vs. ESC) and strategy (BEH vs. COG). These two dimensions are assumed to be overlapping. In other words, in any case of disgust avoidance, both the dimension of focus (in the form of either prevention or escape) and of strategy (either behaviorally or cognitively) are assumed to be present. For example, avoiding to go into a situation which could elicit disgust represents both a focus (here: prevention) and a strategy (here: behavior).

The problems of the 4-factor model we observed in the 4-factor EFA (Study 1) and 4-factor CFA (Study 2; see description above) might have arisen because the subscales of Study 1

measured the overlapping concepts of PREV, ESC, BEH and COG. More specifically, Behavioral Prevention (BP) taps into the constructs of BEH and PRE, Cognitive Prevention (CP) assesses COG and PRE, Behavioral Escape (BE) assesses BEH and ESC, and Cognitive Escape (CE) assesses COG and ESC (see Fig 1C). Re-examining the scale as a whole, we would expect each item of the DAQ to fall on both dimensions of disgust avoidance and thus load on one type of focus (either PREV or ESC) as well as on one type of strategy (either BEH or COG). Based on this, the resulting model (see Fig 1C) would form a 4-factor structure with overlapping factors: PRE (items 1–9), ESC (items 10–17), BEH (items 1–5 + 10–13), and COG (items 6–9 + 14–17). In Study 2 we therefore examined the factor structure of this overlapping 4-factor model, mainly by using *exploratory structural equation modelling* (ESEM; [55]), and its relationship with other constructs. We aimed for a sample size of at least 500 participants, based on Comrey & Lee [56] categorizing a sample size of 500 as 'very good'.

## Relationship with other constructs

We aimed to examine relationships between the DAQ and other instruments aimed at measuring related constructs to evaluate the DAQ's convergent validity. We chose to examine the association between DAQ subscale scores and other disgust-related individual difference measures (disgust propensity & sensitivity) as well as broader emotion-related scales (experiential avoidance & emotion regulation). As we argued earlier, we assume that trait disgust variables, experiential avoidance, and emotion regulation are conceptually related to the construct of disgust avoidance. Although we also emphasized the potential clinical relevance of the DAQ, we did not to include clinical measures yet, because we decided to first focus on the DAQ's construct validity before examining its criterion validity.

People who find the experience of disgust very aversive (heightened disgust sensitivity) would be expected to show a strong tendency to avoid experiencing disgust. Thus, we hypothesized that the subscales of the DAQ would be strongly correlated to a measure of general disgust sensitivity (DPSS–Sensitivity subscale [15]). In addition, people who are disgusted more easily (disgust propensity) are likely to have a higher tendency to avoid, and particularly to prevent, experiencing disgust. We hypothesized that disgust propensity (DPSS–Propensity Subscale; [15]) would be highly associated with PREV, highly to moderately associated with BEH and COG, and moderately associated with ESC. We expected that these predicted relationships between disgust propensity and the DAQ subscales extend to domain-dependent disgust propensity towards pathogen, sexual, and moral disgust propensity (measured with the TDDS; [4]), although likely less pronounced with the more extended disgust domains (i.e., sexual disgust and particularly moral disgust).

Disgust avoidance is assumed to be an individual difference variable that falls under the broad umbrella term of experiential avoidance. We thus hypothesized that the DAQ subscales are moderately associated with experiential avoidance (measured with the BEAQ; [57]). Lastly, we investigated the association of the DAQ with a measure of emotion regulation (measured with the ERQ; [58]). The ERQ consists of one subscale assessing cognitive reappraisal, which refers to a type of antecedent-focused emotion regulation that aims to change the valence of a given situation through cognitive processes. This subscale would be expected to correlate moderately with the PREV and COG subscales of the DAQ. The other ERQ subscale assesses expressive suppression, which is a response focused emotion regulation strategy that aims to control the expression of emotional reaction. This ERQ subscale assesses a component of response-focused emotion regulation that is different from our focus on a person's emotional experience rather than their expression of it. However, we would expect a moderate to low correlation with the ESC subscale.

## Method

**Participants.**   Like in Study 1, we recruited our sample via two university-based partici-pant pools consisting of (Pool 1) first-year bachelor psychology students ($n = 320$; participa-tion in exchange for course credit) and (Pool 2) a broader group of young adults ($n = 193$; participation in exchange for financial compensation: 4€). Participants of Study 1 were excluded from participating in Study 2. The participants (total $N = 513$; 73.7% female) were tested between October 2018 and December 2018. The majority of participants were in their early twenties, either Dutch or German, and studied Psychology (see Table 4 for sample char-acteristics). From the initial $n = 764$ participants, $n = 251$ (32.85%) participants were excluded, because they (a) did not consent to participating/to allowing the use of their data ($n = 76$; 30.28%), (b) did not answer both control questions correctly ($n = 130$; 51.79%), or (c) indicated that they were not motivated enough to properly engage in the study ($n = 45$; 17.93%). The percentage of excluded participants was higher in Study 2 than in Study 1 (15.76%), which might have been due to Study 2 excluding participants based on their motivation, which was not done in Study 1 (i.e., this question was not asked in Study 1).

**Materials.**   The final item set of the *Disgust Avoidance Questionnaire* (DAQ; final item set) consists of 17 items, which are answered on a 7-point Likert scale (1: strongly disagree– 7: strongly agree), and assess people's tendency to avoid experiencing disgust. The DAQ includes four subscales: disgust prevention, disgust escape, cognitive disgust avoidance, and behavioral disgust avoidance. The DAQ as it was presented to the participants is displayed in S2 Appendix (accessible on the OSF: https://osf.io/qnfxg/).

The 16-item *Disgust Propensity and Sensitivity Scale–Revised* (DPSS-R; [15]) assesses gen-eral disgust propensity (i.e., the tendency to experience disgust; 8 items) and disgust sensitivity (i.e., the extent to which the experience of disgust is evaluated as aversive; 8 items). Items are scored on a 5-point scale from 'never' (1) to 'always' (5). The internal consistencies were acceptable (Cronbach's alpha $\alpha = .78$) for the disgust propensity subscale and questionable ($\alpha = .68$) for the disgust sensitivity subscale.

**Table 4. Gender, age, nationalities, and study fields of Study 2 (overall and per recruitment pool).**

|  | Overall ($N = 513$) | Pool 1 ($n = 320$) | Pool 2 ($n = 193$) |
|---|---|---|---|
| Age (Mean, SD)[1] | 21.1 (4.32) | 20.05 (2.15) | 22.78 (6.13) |
| Gender |  |  |  |
| Female | 73.7% | 72.2% | 76.2% |
| Male | 26.3% | 27.8% | 23.8% |
| Nationality |  |  |  |
| Dutch | 25.5% | 20.3% | 34.2% |
| German | 40.5% | 54.1% | 18.1% |
| Other[2] | 33.9% | 25.6% | 47.7% |
| Field of Study |  |  |  |
| Psychology | 76.2% | 99.4% | 37.8% |
| Other[3] | 20.1% | 0.6% | 52.3% |
| Not Studying | 3.7% | 0% | 9.8% |

*Note.*

Pool 1 = participation in exchange for course credit.

Pool 2 = participation in exchange for financial compensation.

[1]Responses were missing for $n = 6$.

[2]Other included a variety of nationalities (e.g., English, Eastern & Southern European, Asian, Baltic, Scandinavian).

[3]Other included a variety of study fields (e.g., Biology, Medicine, Communications, Law, Finance, Economics).

The *Three Domains of Disgust Scale* (*TDDS*; [4]) is a 21-item self-report questionnaire assessing disgust propensity in three domains: moral disgust (e.g., violent behavior), sexual disgust (e.g., incest) and pathogen disgust (e.g., mutilated bodies; spoiled food). Items are rated on a 7-point Likert-scale from 'not at all disgusting' (0) to 'extremely disgusting' (6). The internal consistencies were acceptable ($\alpha$ = .76) for the pathogen subscale, good ($\alpha$ = .83) for the sexual subscale and excellent ($\alpha$ = .90) for the moral subscale.

The *Brief Experiential Avoidance Questionnaire* (BEAQ; [56]) is a brief (15-item) version of the 62-item Multidimensional Experiential Avoidance Questionnaire (MEAQ; [18]). Just like the MEAQ, the BEAQ is a measure of experiential avoidance, which refers to a tendency to avoid experiencing negative emotions, thoughts, memories, and physical sensations [14]. Items are rated on a 6-point Likert scale ranging from 'strongly disagree' (1) to 'strongly agree' (6). The internal consistency was good ($\alpha$ = .82).

The *Emotion Regulation Questionnaire* (ERQ; [57]) is a 10-item self-report scale that assesses people's tendencies to cope with emotions. The ERQ consists of two subscales: Cognitive reappraisal (an antecedent-focused emotion regulation strategy that aims to change the valence of a given situation through cognitive processes) and expressive suppression (a response focused emotion regulation strategy that aims to control the expression of emotional reaction). The items of the ERQ are answered on a 7-point Likert scale ranging from 'strongly disagree' (1) to 'strongly agree' (7). The internal consistencies were good for both subscales ($\alpha$'s = .83 and .82 for the cognitive reappraisal and expressive suppression subscales respectively).

The materials also included the reduced item set (18) of the *Body-related Disgust Avoidance Questionnaire* (B-DAQ) which aims to assess people's tendencies to avoid experiencing body-related disgust, and questionnaires related to the B-DAQ. The B-DAQ is a body-related version of the Disgust Avoidance Questionnaire (DAQ). The B-DAQ and related materials can be found on the OSF (https://osf.io/4mzfs/) and will not be described here because it would extend the scope of the paper.

**Procedure.** Advertisements for the study (Ethics Committee of the University of Groningen Approval code: 18011-SP) were posted on online platforms (Facebook, university-based participant pools), which included a short description of the study and a link that forwarded the participants to the online questionnaires in Qualtrics (Qualtrics, Provo, UT). In Qualtrics, participants were informed about the study (its general aim and content) and asked to give consent to participate in the study and to allow the use of their data for analysis and publication (data usage question). Participants were asked about their demographic information details (age, gender, field of study), filled in the following questionnaires (in that order): DPSS-R, the TDDS, the DAQ, the B- DAQ and related materials (OSF: https://osf.io/4mzfs/), the BEAQ, and the ERQ. We decided on this order because we wanted to present the questionnaires in blocks of similar themes (trait disgust scales, body-related scales, emotion regulation scales). Two control questions were included (one in the DAQ and one in the B-DAQ), which asked participants to select a specific answer category (e.g., please click the left-most answer option) that served as a check to exclude inattentive participants. Lastly, participants were given the possibility to leave notes concerning the questions they just answered and were asked whether they were motivated to participate in the study properly. Participation lasted around 30–45 minutes.

**Analysis.** In the following, we examined the overlapping-factor model. In addition to CFA, we used *Exploratory Structural Equation Modeling* (ESEM; [54]), to investigate the model structure in a more exploratory framework (With ESEM modeling overlap is possible, which is not possible in EFA). Unlike CFA, ESEM does not fix but simply targets cross-loadings to zero. Fixing cross-loadings to zero could potentially be problematic (e.g., providing biased estimates). Despite these shortcomings of the CFA approach, we reported CFA model

statistics and factor loadings for comparability with other studies that focus on the evaluation of instruments' internal structures. With the exception of CFA model statistics and factor loadings, we did not report other CFA-based statistics (factor correlations or CFA-weighted factor scores) because of the potential problems with CFA.

As in Study 1, we considered the following model fit values acceptable: CFI ≥ 0.90, and RMSEA ≤ 0.08. Because of the more complex model structure, we also examined additional fit indices: *The Tucker Lewis Index* (TLI; [59]), and the *Weighted Root Mean Square Residual* (WRMR; [48]), where we considered TLI ≥ 0.95 and WRMR <1.0 acceptable. In case of insufficient model fit, we investigated different modification methods including allowing item error terms to correlate or item reductions. Once a model structure with acceptable fit indices was found we examined the correlations between different subscales using ESEM factor correlations. In addition, we examined the reliability of observed scores by using nonlinear structural equation modeling ([60] in R version 3.6.1 [61] using semTools version 0.5–2 [62], lavaan version 0.6–5 [63]). Lastly, we compared two methods of factor score calculations in order to investigate whether a simple factor score calculation method (unweighted sum scores) yielded similar results to the ESEM-based method (ESEM-weighted sum scores). Unweighted (UW) factor scores were calculated by summing the respective items of each factor. ESEM factor scores were obtained using the 'save = fscores' function in Mplus.

We examined Pearson's correlations between The DAQ subscales (PREV, ESC, BEH, COG) and disgust sensitivity, disgust propensity (general, pathogen, sexual, moral), experiential avoidance, and emotion regulation (cognitive reappraisal, expressive suppression). We assessed each subscale with unweighted (UW) sum scores and ESEM-weighted (ESEM) sum scores (as before, UW factor scores were calculated by summing the respective items of each factor; ESEM factor scores were obtained from Mplus), to further investigate whether using unweighted sum scores is acceptable. We thus reported the correlations of both UW- and ESEM- sum scores of each subscale with the constructs listed above. Because of the low association between UW and ESEM factor scores on the ESC factor, only correlations with ESEM factor scores were interpreted for the ESC factor. We used the Bonferroni correction to adjust for multiple comparisons (64 in total), which led to a corrected alpha value of .0008 per test.

Lastly, we calculated the means of unweighted (UW) sum scores of the DAQ subscales overall and across different demographic variables, including gender, nationality, and field of study. We also reported correlations between DAQ subscales scores and age.

## Results

**Factor structure.** The model fit indices of the CFA model indicated a sufficient fit of the model (CFI = .976, RMSEA = .079 [90% CI: 0.072–0.088], TLI = .966, WRMR = .860). The CFA model with standardized factor loadings can be found in Table 5. In general, most factor loadings were acceptable (>0.4) with the exception of two items on the PREV factor and one item on the ESC factor (<0.3).

The results of the ESEM model were more mixed with an insufficient RMSEA value (.090 [90% CI: 0.082–0.099]), and sufficient CFI (.976), TLI (.956), and WRMR (.695) values. Investigations of ESEM item loadings revealed no item cross-loadings >0.3 (see Table 5), but a number of small target loadings (<0.3) on the PREV (five items) and ESC factor (seven items). PREV was weakly-moderately correlated with ESC and moderately-highly correlated with BEH and COG. ESC had a small negative correlation with BEH (the correlation between ESC and COG was not significant). Lastly, BEH and COG were highly correlated with each other. Because of the RMSEA > .08, the low target loadings in the ESC (and PREV) factor, and the negative correlation between ESC and BEH in the ESEM model, we explored possible

**Table 5. Standardized factor loadings and factor correlations of the CFA and ESEM models (N = 513).**

| Item | CFA | | | | ESEM | | | |
|---|---|---|---|---|---|---|---|---|
| | *PREV* | *ESC* | *BEH* | *COG* | *PREV* | *ESC* | *BEH* | *COG* |
| 1 | **.38** | | **.80** | | .01 | .02 | **.59** | .21 |
| 2 | **.68** | | **.66** | | **.32** | .13 | **.36** | .25 |
| 3 | **.85** | | **.57** | | **.65** | .16 | **.38** | -.02 |
| 4 | **.48** | | **.80** | | .16 | -.06 | **.59** | .16 |
| 5 | **.46** | | **.75** | | .24 | -.16 | **.56** | .08 |
| 6 | .24 | | | **.74** | -.02 | .19 | .17 | **.66** |
| 7 | **.68** | | | **.49** | **.53** | .18 | .04 | **.35** |
| 8 | .25 | | | **.71** | .27 | -.20 | .13 | **.48** |
| 9 | **.34** | | | **.72** | **.43** | -.26 | .03 | **.53** |
| 10 | | **.84** | **1.39** | | -.14 | .13 | **.78** | .07 |
| 11 | | **1.08** | **1.52** | | -.04 | .18 | **.75** | .02 |
| 12 | | **1.56** | **1.75** | | .16 | .17 | **.82** | -.21 |
| 13 | | **1.33** | **1.78** | | .04 | .11 | **1.01** | -.19 |
| 14 | | .26 | | **.90** | -.11 | .10 | -.01 | **.85** |
| 15 | | **.70** | | **1.18** | .08 | **.39** | -.09 | **.87** |
| 16 | | **.61** | | **1.19** | .06 | .26 | .06 | **.79** |
| 17 | | **.51** | | **1.05** | .22 | .01 | -.03 | **.66** |
| *ESEM Factor correlations*[a] | | | | | | | | |
| Prev | | | | | 1.00 | .20 (.004) | .48 | .52 |
| Esc | | | | | | 1.00 | -.05 (.654) | -.14 (.012) |
| Beh | | | | | | | 1.00 | .71 |
| Cog | | | | | | | | 1.00 |

*Note.* Factor loadings of ≥ 0.3 are marked bold.

[a]factor correlations had *p* values < .001, unless otherwise specified in brackets.

adjustments to the model through item exclusions or allowing item errors to correlate. Because the different methods we explored did not yield adjusted models that performed better with regards to the issues named above (RMSEA's > .08/low target loadings remained/negative correlations increased), we decided not to include any adjustments to the model structure.

**Reliabilities and factor scores.** With regard to the factor reliabilities, we found the following non-linear SEM reliabilities for the separate factors: PREV: .93, ESC: .97, BEH: .93, COG: .92, indicating high factor reliability in all factors. The correlation between UW and ESEM factor scores was very high for the BEH ($r(511) = .97$, p < .001) and the COG factor ($r(511) = .96$, p < .001), high for the PREV factor ($r(511) = .83$, p < .001), but small for the ESC factor ($r(511) = .18$, p < .001). The unweighted scores of the BEH, COG, and PREV factors, although a little less accurate, will (considering the correlations listed above) probably serve their intended purpose. However, this does not apply to the ESC factor.

**Relationships with other constructs.** The distributions of the variables showed no severe deviations from the normal distribution. The results of the correlation analyses can be found in Table 6. P values > .0008 are given in brackets. Disgust sensitivity had moderate-high associations with PREV, BEH, and COG ($r$'s around .40), which is partly supporting our predictions. However, no statistically significant association of disgust sensitivity and ESC (ESEM) was found. PREV, BEH, and COG showed moderate-high associations with general disgust propensity ($r$'s around .45 - .50) and pathogen disgust propensity ($r$'s around .40), and moderate associations with sexual disgust propensity ($r$'s around .30). The correlations of DAQ

**Table 6. Correlations of the DAQ subscales with other constructs per score calculation method (unweighted and ESEM-based sum scores; *N* = 513).**

| | UW | | | | ESEM | | | |
|---|---|---|---|---|---|---|---|---|
| | *PREV* | *ESC* | *BEH* | *COG* | *PREV* | *ESC* | *BEH* | *COG* |
| Disgust Sensitivity | .43 | .41 | .40 | .41 | .39 | .07 (.102) | .38 | .39 |
| Disgust Propensity | | | | | | | | |
| General | .50 | .47 | .50 | .45 | .36 | -.01 (.905) | .50 | .49 |
| Pathogen | .40 | .42 | .43 | .37 | .26 | .02 (.655) | .43 | .39 |
| Sexual | .29 | .31 | .31 | .27 | .17 | .03 (.575) | .29 | .28 |
| Moral | .10 (.028) | .04 (.380) | .07 (.111) | .07 (.133) | .14 (.001) | .02 (.728) | .05 (.259) | .05 (.312) |
| Experiential Avoidance[a] | .35 | .32 | .30 | .35 | .40 | .12 | .27 | .29 |
| Emotion Regulation | | | | | | | | |
| Cognitive Reappraisal | .19 | .19 | .16 | .22 | .11 (.013) | -.04 (.434) | .17 | .25 |
| Expressive Suppression | .06 (.175) | .03 (.516) | -.01 (.858) | .10 (.022) | .12 (.008) | .07 (.095) | -.00 (.975) | .08 (.069) |

*Note*. All *p* values < .0008 (α adjusted for multiple comparisons) unless otherwise specified in brackets. Disgust Sensitivity: DPSS-R Sensitivity Subscale. Disgust Propensity (General): DPSS-R Propensity Subscale. Pathogen, Sexual, & Moral Disgust Propensity: TDSS subscales. Experiential avoidance: BEAQ. Cognitive Reappraisal & Expressive Suppression: ERQ subscales. UW = unweighted sum scores; ESEM = ESEM-based sum scores.

[a] Items of the MEAQ served as a basis for the type and wording of DAQ items. Because the BEAQ is a short version of the MEAQ, there is a similarity between two BEAQ and DAQ items (*When unpleasant memories come to me, I try to put them out of my mind* [BEAQ] ~ *When I think about something gross, I push those thoughts out of my mind* [DAQ]; *I am quick to leave any situation that makes me feel uneasy* [BEAQ] ~ *I am quick to leave any situation that makes me feel disgusted* [DAQ]).

subscales with moral disgust propensity were not significant (*p*'s > .0008). ESC (ESEM) did not show statistically significant associations with any measure of disgust propensity. In line with our prediction, the associations of the DAQ subscales with disgust propensity were weaker for more extended disgust domains (sexual and especially moral disgust propensity). The associations of experiential avoidance with PREV, BEH, and COG were of moderate size (*r*'s around .30), which is in line with our predictions. However, ESC (ESEM) only showed a weak association with experiential avoidance. In line with our predictions, the cognitive reappraisal subscale of the emotion regulation questionnaire was moderately correlated with COG (*r*'s around .23), but not consistently related to PREV (weak correlation with UW only). Against our prediction, we did not find a significant association between expressive suppression and ESC.

In sum, the results were partly in line with our predictions. The correlations of the investigated variables with the PREV, BEH, and COG factors, although often a bit less strong than predicted, generally supported our hypotheses. In general, associations were comparable across UW and ESEM for the PREV, BEH, and COG factors, although UW scores often provided slightly higher estimates than ESEM scores. For the ESC factor, UW and ESEM correlations showed very low correspondence. The ESEM scores of the ESC factor were not found to be correlated with most constructs we investigated.

**Means of DAQ unweighted sum scores.** The means of unweighted sum scores of the DAQ subscales overall and across different demographic variables (gender, nationality, & field of study) and correlations of unweighted sum scores of the DAQ subscales with age can be found in Table 7.

## General discussion

The aim of the project was to develop a questionnaire that assesses people's tendency to avoid experiencing disgust, with a specific aim on distinguishing between prevention- and escape-focused forms of disgust avoidance. A pool of potential items (25), was condensed in a stepwise

**Table 7. Means and standard deviations of DAQ unweighted sum scores (overall and per gender, nationality, & field of study) and correlations with age.**

|  | N | PREV | ESC | BEH | COG |
|---|---|---|---|---|---|
| Overall | 513 | 44.07 (9.52) | 39.28 (8.01) | 43.60 (9.37) | 39.74 (8.68) |
| Gender |  |  |  |  |  |
| Male | 135 | 40.17 (10.26) | 34.96 (8.80) | 39.53 (10.37) | 35.59 (9.39) |
| Female | 378 | 45.63 (8.84) | 40.82 (7.10) | 45.06 (8.54) | 41.22 (7.09) |
| Nationality |  |  |  |  |  |
| Dutch | 131 | 42.43 (9.10) | 38.44 (7.24) | 41.80 (8.81) | 39.07 (8.20) |
| German | 208 | 43.95 (9.30) | 38.77 (7.85) | 43.30 (9.27) | 39.42 (8.38) |
| Other[1] | 174 | 45.45 (9.91) | 40.51 (8.62) | 45.32 (9.65) | 40.64 (9.32) |
| Field of Study |  |  |  |  |  |
| Psychology | 391 | 43.86 (9.49) | 39.16 (7.95) | 43.53 (9.28) | 39.49 (8.65) |
| Other[2] | 103 | 45.55 (9.32) | 40.01 (8.33) | 44.50 (9.74) | 41.07 (8.47) |
| Not Studying | 19 | 40.21 (10.07) | 37.79 (7.47) | 40.21 (8.71) | 37.79 (9.87) |
| *Correlations* |  |  |  |  |  |
| Age[3] | 507 | -.05 ($p$ = .248) | -.03 ($p$ = .515) | -.03 ($p$ = .506) | -.05 ($p$ = .251) |

[1]Other included a variety of nationalities (e.g., English, Eastern & Southern European, Asian, Baltic, Scandinavian).

[2]Other included a variety of study fields (e.g., Biology, Medicine, Communications, Law, Finance, Economics).

[3]Responses were missing for $n$ = 6; Reported correlations are Pearson's correlation coefficients.

item reduction extracting single- and multi-factor EFA models. Based on a 4-subscale distinction (behavioral disgust prevention, cognitive disgust prevention, behavioral disgust escape, and cognitive disgust escape), we excluded a total of 8 items, resulting in a reduced set of 17 items. Evaluation of this 4-factor model using CFA in a new sample showed a problematic model fit. A conceptual re-evaluation of the model structure led us to specify an adapted model to incorporate the conceptual overlap between two dimensions of disgust avoidance: focus (prevention vs. escape) and strategy (behavioral avoidance vs. cognitive avoidance). In the new model, we allowed each item to load on one type of dimension (either disgust prevention or disgust escape) AND one type of strategy (either behavioral or cognitive disgust avoidance). After evaluation of this overlapping 4-factor model using CFA and ESEM, we examined inter-factor correlations, reliability of observed scores, factor score calculation methods, and correlations with existing disgust-related and broader emotion-related measures. In general, we observed promising results for the factors disgust prevention, behavioral disgust avoidance, and cognitive disgust avoidance, but not for the disgust escape factor.

### Disgust avoidance questionnaire

The fit of the DAQ was acceptable (CFI ≥ 0.95, RMSEA ≤ 0.08, WRMR <1.0) when evaluated in a CFA framework. When evaluating the model using ESEM, three of our fit indices also indicated acceptable model fit (CFI ≥ 0.95, TLI ≥ 0.95, WRMR <1.0), but one indicated a non-acceptable model fit (RMSEA = 0.09). We therefore evaluated the model fit as promising but in need of further investigation. Items generally showed to load well on their presumed factors, with the exception of the escape factor in the ESEM framework. Disgust prevention, behavioral disgust avoidance, and cognitive disgust avoidance correlated moderately to highly with each other (see Table 5 for inter-factor correlations), indicating that the DAQ subscales (except for the escape factor) represent related (but distinct) constructs. Individual difference in disgust prevention, behavioral disgust avoidance, and cognitive disgust avoidance are thus likely to be closely related, but may be of differential relevance to other psychological constructs.

The DAQ (except for the escape factor) showed good convergent validity. In line with predictions, participants who find the experience of disgust more aversive (disgust sensitivity) were also more likely to report that they want to prevent experiencing disgust and engage in disgust-avoiding behaviors and cognitions. The same was found for participants who are easily disgusted (disgust propensity), particularly by pathogen-relevant stimuli and, to a lesser extent, by sexual stimuli. Although we expected weak correlations in more extended disgust domains, it was surprising that moral disgust was not found to be significantly correlated with any subscale of the DAQ. It might be that moral situations/behaviors are not as strong as sexual/pathogen stimuli in eliciting disgust, thus only resulting in small associations to domain-independent measures of disgust (e.g., the DAQ). In addition, it might also be that when answering general questions about disgust avoidance, people may rather have quite concrete disgust elicitors in mind, which might not readily include morally disgusting situations. Due to the high number of comparisons, our power to detect weak relationships was low, meaning that there was an increased chance of false negatives. In line with our predictions, we found that participants who are likely to avoid negative emotions or thoughts (experiential avoidance), were also more likely to report that they want to prevent experiencing disgust and engage in disgust-avoiding behaviors and cognitions.

In general, the correlations of disgust avoidance with disgust sensitivity, disgust propensity, and experiential avoidance (the correlation between BEAQ and DAQ might have been slightly inflated due to the overlap between two items of the BEAQ and the DAQ), which have been implicated in a number of mental disorders [e.g., 6,38], can be taken to reflect adequate convergent validity. On a theoretical level, a strong tendency to avoid experiencing disgust, especially in combination with a heightened propensity to experience disgust, might contribute to the detrimental coping mechanisms observable in several disgust-relevant mental disorders (e.g., performing rituals in OCD, extreme dieting in eating disorders, avoidance of intercourse in sexual disorders, isolation in PTSD). Assessing disgust avoidance might help understand the drive behind maladaptive coping mechanisms, and as a result provide new directions for treatment. For that reason, it is crucial for future research to establish the DAQ's relationship with and relevance to clinical constructs.

**Disgust prevention and escape.**   The disgust prevention subscale consists of nine items that aim to assess people's tendency to prevent experiencing disgust. When evaluated in combination with the other subscales using CFA and ESEM approaches some items showed poor target loadings (CFA: 1 item, ESEM: 5 items). The prevention scale showed high reliability and correlated moderately to highly with the other subscales. Comparing ESEM-weighted factor scores with unweighted factor scores displayed a good correspondence, and both were moderately-highly related to disgust sensitivity and propensity (general, pathogen, and moderately to sexual) and moderately related to experiential avoidance, indicating that our measure of disgust prevention was associated with conceptually related constructs. An important next step would be to examine how disgust prevention (which we assume to reflect a more deliberate process) contributes to strategic avoidance behaviors/cognitions.

The disgust escape subscale consists of eight items that aim to assess people's tendency to escape from the experience of disgust. Although the scale displayed high reliability, and CFA model results indicated mostly well-loading items, the ESEM model showed poor loadings on seven items. Surprisingly, disgust escape was negatively correlated with behavioral disgust avoidance. ESEM and unweighted factor scores did not correspond, and ESEM factor scores were not found to be correlated with most conceptually-related constructs. Due to these problematic findings related to the escape subscale, we concluded that the DAQ was not able to measure people's tendency to escape from the experience of disgust. A possible reason for the poor performance of the escape subscale could be that our form of assessment (self-report)

may have been unsuitable to assess the concept of disgust escape. We theorized disgust escape to be a reactive and rather automatic form of disgust avoidance, compared to the more strategic disgust prevention. Such an automatic process may be less accessible through conscious reflection (i.e., self-report questionnaire; e.g., [64]). Future investigations are needed to establish a suitable tool for measuring disgust escape. For example, exposing people to feelings of disgust and measuring their tendency to escape from that feeling or measuring implicit/automatic association of feeling disgusted with escape (e.g., via an Implicit Association Test; [65]) may be more suitable assessment options. When designing such a task, it would be important to include both strategies of avoidance (i.e., cognition and behavior). Development of an appropriate assessment tool of disgust escape would make it possible for future research to examine how the roles of disgust escape and disgust prevention in disgust-relevant mental disorders can be differentiated.

**Behavioral and cognitive disgust avoidance.** The behavior subscale consists of nine items that aim to assess people's tendency to engage in disgust-avoiding behavior. In both the CFA and ESEM models, items had good (and some acceptable) target loadings. The cognition subscale consists of eight items that aim to assess people's tendency to engage in cognitive avoidance of disgust. CFA and ESEM models showed items with good target loadings. Both scales showed high reliability, a close correspondence between ESEM and unweighted factor scores, and moderate-high correlations with conceptually related constructs. The behavioral and cognitive subscales were highly correlated with each other and showed similar correlations with other constructs (see Tables 5 and 6). Although some minor differences in the strength of the correlations may be observed (e.g., with cognitive reappraisal), future research is needed to examine the distinction between the two subscales. On a theoretical level, distinguishing between behavioral and cognitive avoidance might provide insights into engagement in more overt (i.e., behaviors) vs. covert (cognitions) avoidance. Those types of avoidance might be related to distinct predictors (e.g., certain traits or contexts) and outcomes (e.g., clinical variables). Especially in a therapeutic context (e.g., exposure therapy), distinguishing between behavioral and cognitive disgust avoidance may be of relevance to tailor treatment focus. These theoretical arguments need to be examined in future research establishing the (distinctive) role of behavioral and cognitive avoidance in (the treatment of) psychopathology.

## Limitations and recommendations for future research

This project represents a first step into making the concept of disgust avoidance assessable in form of a self-report questionnaire. We want to emphasize that there are several limitations that should be considered in future use of the scale. Most importantly, the nature of the project was exploratory. This is due to the adaptation of the internal structure of the scale throughout this project starting at our initial assumption of a two-factor structure (Prevention & Escape), to splitting that two-factor structure into four non-overlapping factors (Behavioral Prevention, Cognitive Prevention, Behavioral Escape, Cognitive Escape), and ultimately modeling an overlapping four-factor structure (Disgust Prevention, Disgust Escape, Behavioral Disgust Avoidance, Cognitive Disgust Avoidance). Although we did not realize this from the start, we considered the overlapping four-factor model to be most representative of the internal structure of the scale. In general, model fit statistics seemed to indicate an acceptable fit of the overlapping four-factor model, with the exception of the RMSEA value of the ESEM model. RMSEA can lead to model rejection in case of high reliability and small specific variance of the variables/items, even if there is only minor model error [52]. We therefore evaluate our model to have a promising fit, with possible minor model error.

Due to the limitations of the project, we want to emphasize that the scale should be subjected to scrutiny through future research. With regard to the use of the scale in research, unweighted sum scores may be used to calculate scores on the prevention, behavior, and cognitive subscales, although caution should be taken because unweighted sum scores may provide slightly overestimated parameters. This particularly applies to the disgust prevention subscale, which showed some weak-loading items. The problems associated with the escape subscale (low target loadings, negative/non-significant correlations with other factors, and conceptually-related constructs) imply that the escape subscale should not be used (in its current form) in future research. As was argued earlier, self-report may not be a suitable form to assess disgust escape. Although we certainly welcome attempts to refine the escape subscale (in its self-report form), at this point we believe it may be necessary to use a different operationalization that is not as dependent on reflective processes (see above).

We specifically recommend the use of the three DAQ subscales in future research with the goal to investigate their distinctive relevance to psychological constructs and processes. Although the obtained inter-factor correlations and factor loadings suggest conceptually distinctive constructs, our correlational analysis did not reveal clear differences in the correlations between the subscales (exception: escape subscale) and the limited number of investigated constructs. Future research is needed to examine the distinctive roles of the DAQ subscales in a larger number of psychological constructs/processes. In order to establish the clinical relevance of the DAQ, we need research examining the role of the DAQ subscales in processes relevant to disgust-related symptomatology (e.g., eating disorders, sexual disorders, anxiety disorders).

Because we used relatively restricted samples in the current project (consisting of mainly white, highly educated, and relatively young people), validation of the DAQ in broader/different samples is needed to establish its general applicability. Although we combined both judgmental and statistical criteria (e.g., in our item selection in Study 1) in order to decrease the sample-dependency of our results, it remains to be examined whether our results can be replicated in different samples. We also would like to acknowledge that a substantial number of participants had to be excluded from our studies (particularly from Study 2) and that a selection bias cannot be ruled out. In the meantime, this also points to the importance of including control questions (as was done in the current studies) to improve the quality of the data and to limit noise related to random answers of non-motivated participants. Lastly, the sample sizes of our studies were based on general rules of thumb instead of a-priori power analyses. Although we consider the general rules of thumb to be sufficient due to the relatively low complexity of our models, we recommend future research on the DAQ to conduct a-priori power analyses.

## Conclusion

The current studies represent a critical first step towards examining the concept of disgust avoidance. We developed a questionnaire assessing people's tendency to avoid experiencing disgust (DAQ) across two overlapping dimensions, namely focus (prevention vs. escape) and strategy (behavioral avoidance vs. cognitive avoidance). The results generally seem promising for three of the DAQ subscales (disgust prevention, behavioral disgust avoidance, cognitive disgust avoidance) but made us question the suitability of self-report to assess disgust escape. Future research is needed to explore alternative methods to measure disgust escape and to examine the distinctive role of the DAQ subscales in other psychological constructs and processes.

## Supporting information

**S1 Table. Items of the MEAQ, EAQ, and CAQ used as source items for the DAQ.**
(DOCX)

**S2 Table. Two-factor EFA on the prevention-focused items (initial item set).**
(DOCX)

**S3 Table. Two-factor EFA on the escape-focused items (initial item set).**
(DOCX)

**S1 Appendix. The initial item set (25) of the Disgust Avoidance Questionnaire (DAQ).**
(DOCX)

**S2 Appendix. The reduced item set (17) and subscale calculation of the Disgust Avoidance Questionnaire (DAQ).**
(DOCX)

## Acknowledgments

We would like to thank Prof. Dr. Marieke Timmerman for her advice on the stepwise item reduction method.

## Author Contributions

**Conceptualization:** Paula von Spreckelsen, Nienke C. Jonker, Ineke Wessel, Klaske A. Glashouwer, Peter J. de Jong.

**Data curation:** Paula von Spreckelsen.

**Formal analysis:** Paula von Spreckelsen, Nienke C. Jonker, Jorien Vugteveen.

**Investigation:** Paula von Spreckelsen.

**Methodology:** Paula von Spreckelsen, Nienke C. Jonker, Jorien Vugteveen, Ineke Wessel, Klaske A. Glashouwer, Peter J. de Jong.

**Project administration:** Paula von Spreckelsen.

**Software:** Paula von Spreckelsen, Nienke C. Jonker, Jorien Vugteveen.

**Supervision:** Ineke Wessel, Klaske A. Glashouwer, Peter J. de Jong.

**Visualization:** Paula von Spreckelsen.

**Writing – original draft:** Paula von Spreckelsen.

**Writing – review & editing:** Paula von Spreckelsen, Nienke C. Jonker, Jorien Vugteveen, Ineke Wessel, Klaske A. Glashouwer, Peter J. de Jong.

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
