## [Decision Letter · Decision Letter 0]

3 Sep 2020

PONE-D-20-16240

Individual Differences in Avoiding Feelings of Disgust: Development and Construct Validity of the Disgust Avoidance Questionnaire

PLOS ONE

Dear Dr. von Spreckelsen,

Thank you for submitting your manuscript to PLOS ONE. After careful consideration, we feel that it has merit but does not fully meet PLOS ONE’s publication criteria as it currently stands. Therefore, we invite you to submit a revised version of the manuscript that addresses the points raised during the review process.

**Although the two Reviewers appreciated the writing style of the article, the idea behind the disgust avoidance construct, the data analyses and their interpretation, however, they also highlighted important theoretical (Introduction) and methodological shortcomings that require careful revision by the Authors in order for the manuscript to be suitable for publication. I therefore suggest that the Authors take the numerous comments of the Reviewers seriously and proceed addressing all of them.**

We look forward to receiving your revised manuscript.

Kind regards,

Stefano Federici, Ph.D.

Academic Editor

PLOS ONE

Journal Requirements:

2.We note that you have stated that you will provide repository information for your data at acceptance. Should your manuscript be accepted for publication, we will hold it until you provide the relevant accession numbers or DOIs necessary to access your data. If you wish to make changes to your Data Availability statement, please describe these changes in your cover letter and we will update your Data Availability statement to reflect the information you provide.

3. Your abstract cannot contain citations. Please only include citations in the body text of the manuscript, and ensure that they remain in ascending numerical order on first mention

Additional Editor Comments (if provided):

Although the two Reviewers appreciated the writing style of the article, the idea behind the disgust avoidance construct, the data analyses and their interpretation, however, they highlighted important theoretical (Introduction) and methodological shortcomings that require careful revision by the Authors in order for the manuscript to be suitable for publication. I therefore suggest that the Authors take the numerous comments of the Reviewers seriously and proceed addressing all of them.

Reviewers' comments:

Reviewer's Responses to Questions

**Comments to the Author**

1. Is the manuscript technically sound, and do the data support the conclusions?

Reviewer #1: Yes

Reviewer #2: Partly

2. Has the statistical analysis been performed appropriately and rigorously? 

Reviewer #1: Yes

Reviewer #2: No

3. Have the authors made all data underlying the findings in their manuscript fully available?

Reviewer #1: Yes

Reviewer #2: Yes

4. Is the manuscript presented in an intelligible fashion and written in standard English?

Reviewer #1: Yes

Reviewer #2: Yes

5. Review Comments to the Author

Reviewer #1: I have now completed a review of the manuscript "Individual differences in avoiding feelings of disgust: development and construct validity of the Disgust Avoidance Questionnaire". In a scientifically rigorous, logically structured, comprehensive style, the authors present their work focused on development and validation of a new psychometric tool that would measure an original construct of disgust avoidance. Although the Introduction does not present so much of the current state of knowledge (see below), the authors set out clear hypotheses and especially the way they present their methodological/statistical approach to test these is very detailed. They conclude that disgust avoidance might be composed of four partly overlapping factors, although one of them (escape avoidance) showed a poor fit to the model. Finally, acknowledging some of the limitations of the study, they outline potential applicability of the construct and directions for future research. Overall, the study is based on a fairly good sample size (but see below), the data are well analysed and results correctly interpreted. The text is written in very good English. Therefore, I think the manuscript would deserve a publication.

However, I also see some major issues which should be addressed prior to acceptance:

1) Indeed, the idea of disgust avoidance is interesting. However, having read the manuscript, I am still not convinced of its relevance and especially impact into practice. I mean it is an inherent characteristic of disgust that the individual will try to avoid it. As the authors correctly argue, higher disgust propensity and sensitivity should ultimately lead to higher disgust avoidance, so developing a new construct and measuring disgust avoidance might seem redundant. Having said that, I am really not sure how this could help us in therapeutic intervention for people suffering from mental disorders associated with dysregulated disgust, e.g. OCD, certain phobias, behavioural disorders, etc. Perhaps authors could be more specific on this point and stress more the importance of knowing one’s disgust avoidance. Do we really need it? I would need more persuasion. I would also like to see a better argument for a tendency of some people seeking disgust experiences and enjoying them. This works for fear, but disgust - I’m not so certain.

2) The introduction is very brief and goes straight to the point of disgust avoidance, which unfortunately, leads to some conceptual inaccuracies or shortcomings (e.g., in the first sentence, the authors claim “Disgust is a basic emotion that is ingrained in all of us.”, which, without any reference or more details, is oversimplified and speculative, because the concept of basic emotions is one of many). Disgust is a very complex emotion and has gone through massive theoretical and experimental development in the last decades. While I admit that there are many reviews and book chapters more suitable for that and an introduction of a research paper should stay concise, I would still like to see the authors more reflecting on existing findings related to disgust. Regretfully, I must say that this is also associated with an exceptionally low number of references (37 for the whole MS, 17 for the Introduction). It means the authors have neglected a great deal of recent literature.

3) It is not clear to me, why the authors give so much attention to CFA of the model of four overlapping factors, if they correctly state that CFA is not an appropriate method for analysing inter-correlated factors. I understand the argument of comparability with other studies, but then the whole section on the CFA could be extensively shortened.

4) Given the nature of data collection (online study), the sample size seems a bit low. The authors should justify the sample size by conducting a power-analysis prior to the study. Also, they should report effect sizes for all their results.

5) I did not understand why the authors ran EFA and CFA separately for each factor and then for the four-factor model. I am not sure if this is a correct use of the method. Usually, factor analyses are run for the whole model (in this case with four factors), not separately for individual factors.

6) Throughout the whole manuscript, I did not find any results concerning mean sum scores of the DAQ, only factor scores are being reported. I think this should be included in the manuscript. The authors should also analyse and report how the main individual characteristics (gender, age, education) affect the DAQ scores. Unfortunately, this is also missing.

I also have some specific minor issues/comments:

1) Line 35-36: The sentence is grammatically incorrect and does not make sense, please revise.

2) Line 50-53: The list of disgust elicitors is much longer and the authors should provide more examples, e.g. small animals, political orientation,… and describe also magical thinking in relation to disgust.

3) Line 64-67: This sentence is just repeating what has been already said above (line 56-60). Also, please change “Research so far has identified,…” to “Research has so far identified…”.

4) Line 113-114: This sentence is a redundant as it is again a repetition of the previous one (e.g. line 85-86).

5) Line 114-117: It is not correct to say that “pathogen disgust evolved to protect humans from disease-inflicting stimuli that cannot be seen or otherwise detected…”. If this had been the case, disgust could have been triggered by any stimulus. In fact, we can guess fairly well on the presence of pathogens, even if we cannot see them (rotten food, worms, insect, sick people, bodily fluids,…).

6) Line 136-137: Again, this sentence is not totally correct. Contamination is only one threat disgust responds to (next to disease transmission, intoxication,…).

7) Line 152: should be ‘help clarify’, please revise

8) Line 160: please delete “promotes”

9) Line 175: Unless it is a requirement of PLoS One, I find the structure here very unusual. Why is ‘Method’ comprised of ‘Participants’ and separated from ‘Materials’? Generally, you should have a chapter ‘Material and Methods’ where ‘Participants’ should be the first subchapter followed by ‘Assessment’, ‘Procedure’, ‘Statistical Analysis’,…

10) It is not clear to me who was included in Sample 2.

11) Line 212: please add ‘to’ after ‘(1)’

12) Line 247-248: This is the first time the authors mention ‘a subset of a domain-specific version of the DAQ’, but the reader has no idea, what it is. It should have been explained earlier with the description of the psychometrics used in the study.

13) Line 296, Analysis Plan: Why did the authors 13) Line 303: I am not sure what the authors mean by ‘unidimensional factors’. I think factors should always be unidimensional.

14) Line 494: Wouldn’t a Spearman correlation be better given the fact that the distribution of DAQ scores should deviate from normality as they are based on a Likert-scale items (as the authors correctly acknowledged earlier)? I also did not understand if the authors correlated sum scores or factor scores of the DAQ and other measures, it is not very clear from the text.

Reviewer #2: The goal of the present paper was to develop and initially validate a self-report measure of disgust avoidance. Overall, the paper is well-written and the idea behind the disgust avoidance construct is intriguing. The paper would benefit from a stronger justification for the development of the scale and need for this measure in the assessment and treatment of psychological disorders. There are also a number of analytic ambiguities that make it difficult to assess the quality of this measure. I’m not convinced of the factor structure and final measure. Below, I outline my concerns in more detail.

The authors appropriately raise the similarity between experiential avoidance and disgust avoidance, particularly with regard to psychopathology. Is disgust avoidance just a specific form of experiential avoidance? As a relatively large body of research has linked experiential avoidance with various psychological disorders (all those mentioned in connection with disgust sensitivity), why is a disgust avoidance measure necessary? Why isn’t experiential avoidance sufficient? Although disgust has been related to a number of psychological disorders, I don’t think that it is fair to say that disgust avoidance is the only or primary form of avoidance underlying these disorders. It would be useful to distinguish these constructs further and provide justification for the new measure.

How exactly was the “broader sample of young adults” recruited?

A large proportion of Sample 2 (roughly a third, 251 out of 764) was excluded. How many fell into each reason for exclusion? What might explain the higher exclusion rate for Sample 2 compared to Sample 1? Did excluded participants differ from included participants with regard to demographics or any primary study variables?

How was sample size determined?

Given the claims in the intro that a measure of disgust avoidance would be useful for clinical purposes, it is strange that no measures of psychopathology were included. What was the reasoning behind the validity measures that were chosen and the exclusion of mental health measures?

What was the rationale for the questionnaire order in Sample 2? Any concerns about order effects?

Were there any differences between participants based on recruitment method?

For Sample 1, were separate EFAs conducted with the a priori subscale items or was EFA conducted with all items? Based on Table 2, it seems like an EFA was conducted with all of the items, comparing a single and a four factor solution. Why not the proposed two-factor solution? However, the write-up makes it sound like separate EFAs were conducted with each set of subscale items. There needs to be clarification as to what analyses were exactly conducted.

p. 15 - It is very confusing when item numbers are referred to in-text, but those numbers do match item numbers in tables (e.g., Table 2). Keep item numbers consistent or provide item wording. Also, based on Table 2, it is unclear why certain items were retained when they seem to cross-load on factors or load weakly (e.g., item 4). Same issue with Table 3.

p.18 – the shift in labels for the 4 factors is jarring and unclear. More explanation and justification for this change needs to be provided.

6. PLOS authors have the option to publish the peer review history of their article (what does this mean?). If published, this will include your full peer review and any attached files.

Reviewer #1: No

Reviewer #2: No

---

## [Author Response · Author response to Decision Letter 0]

10 Nov 2020

Dear Prof. Federici, 

On behalf of all authors, I hereby submit our revised manuscript PONE- D-20-16240 "Individual Differences in Avoiding Feelings of Disgust: Development and Construct Validity of the Disgust Avoidance Questionnaire". We would like to thank you as well as the reviewers for evaluating our manuscript.

We appreciate the constructive and stimulating comments, as well as the time and effort you and the reviewers invested in reading the manuscript and suggesting improvements. We carefully revised the manuscript and addressed each issue raised by the reviewers. 

Based on the reviewer’s comments, we extended our theoretical background (e.g., by incorporating more recent literature; by elaborating on arguments) and made some structural changes to the manuscript (by separately discussing Study 1 and Study 2; by revising tables) with the goal to improve comprehensibility of the paper. We adapted and added methodological details (e.g., presentation of sample characteristics; clarification of analytic strategies) and statistical details (e.g., revised presentation of Study 1 results; reporting of DAQ mean scores), and raised concerns pointed out by the reviewers in the discussion (e.g., power analysis). 

Below we describe in detail how we handled each of the comments. Each comment is literally quoted, followed by our reply. Line references to the changes made to the manuscript refer to the clean manuscript (in which all track changes were accepted). 

We hope that you will consider the comments to be adequately addressed and find the revised manuscript suitable for publication in PLOS ONE.

Sincerely, also on behalf of the other authors,

Paula von Spreckelsen

Reviewer 1 

I have now completed a review of the manuscript "Individual differences in avoiding feelings of disgust: development and construct validity of the Disgust Avoidance Questionnaire". In a scientifically rigorous, logically structured, comprehensive style, the authors present their work focused on development and validation of a new psychometric tool that would measure an original construct of disgust avoidance. Although the Introduction does not present so much of the current state of knowledge (see below), the authors set out clear hypotheses and especially the way they present their methodological/statistical approach to test these is very detailed. They conclude that disgust avoidance might be composed of four partly overlapping factors, although one of them (escape avoidance) showed a poor fit to the model. Finally, acknowledging some of the limitations of the study, they outline potential applicability of the construct and directions for future research. Overall, the study is based on a fairly good sample size (but see below), the data are well analysed and results correctly interpreted. The text is written in very good English. Therefore, I think the manuscript would deserve a publication. However, I also see some major issues which should be addressed prior to acceptance:

We would like to thank the first reviewer for taking the time and effort to evaluate our manuscript. We were happy to read the reviewer’s positive evaluation of our manuscript and appreciate the reviewer’s concerns. In the following, we hope we were able to address the reviewer’s comments adequately. Please note that line references to the changes made to the manuscript refer to the clean manuscript (in which all track changes were accepted).

1) Indeed, the idea of disgust avoidance is interesting. However, having read the manuscript, I am still not convinced of its relevance and especially impact into practice. I mean it is an inherent characteristic of disgust that the individual will try to avoid it. As the authors correctly argue, higher disgust propensity and sensitivity should ultimately lead to higher disgust avoidance, so developing a new construct and measuring disgust avoidance might seem redundant. Having said that, I am really not sure how this could help us in therapeutic intervention for people suffering from mental disorders associated with dysregulated disgust, e.g. OCD, certain phobias, behavioural disorders, etc. Perhaps authors could be more specific on this point and stress more the importance of knowing one’s disgust avoidance. Do we really need it? I would need more persuasion. I would also like to see a better argument for a tendency of some people seeking disgust experiences and enjoying them. This works for fear, but disgust - I’m not so certain.

In accordance with the reviewer’s comments, we adapted the introduction at several points. Due to the number and the extent of changes to the introduction, we decided not to copy-paste the changes into the response letter, but to merely refer to lines in the manuscript. 

We clarified the relevance of disgust avoidance. More specifically, we emphasized the distinction between disgust avoidance and disgust propensity/sensitivity and addressed research investigating the relationship between disgust avoidance and disgust propensity/sensitivity (lines 92 – 105). 

We emphasized the role of avoidance in the persistence of disgust associations (lines 106 – 120). In addition, we extended our argumentation of the relevance of disgust avoidance to psychopathology (by using the example of OCD; lines 121 -133). Furthermore, we discuss previous research assessing avoidance of disgust/disgusting stimuli (lines 202 – 212), which may be considered as a sign that there is a research interest in assessing the concept of disgust avoidance.

We also provided more explanation and research on the argument that the experience of disgust can be experienced as enjoyable (lines 85 – 91).

2) The introduction is very brief and goes straight to the point of disgust avoidance, which unfortunately, leads to some conceptual inaccuracies or shortcomings (e.g., in the first sentence, the authors claim “Disgust is a basic emotion that is ingrained in all of us.”, which, without any reference or more details, is oversimplified and speculative, because the concept of basic emotions is one of many). Disgust is a very complex emotion and has gone through massive theoretical and experimental development in the last decades. While I admit that there are many reviews and book chapters more suitable for that and an introduction of a research paper should stay concise, I would still like to see the authors more reflecting on existing findings related to disgust. Regretfully, I must say that this is also associated with an exceptionally low number of references (37 for the whole MS, 17 for the Introduction). It means the authors have neglected a great deal of recent literature.

We thank the reviewer for addressing these issues. We deleted the first sentence of the introduction and extended the theoretical background/introduction of the study by discussing:

- the universality/context-dependence of disgust elicitors (lines 55 – 57)

- the role of disgust at a broader societal level (lines 63 - 64)

- publication trends on disgust (lines 65 – 66)

- the relationships between disgust propensity/sensitivity and disgust avoidance (lines 73 – 78 & 92 – 105)

- the potential appetitive quality of experiencing disgust (lines 85 – 91)

- the persistence of disgust associations (lines 106 – 120)

- the potentially distinct role of disgust avoidance to psychopathology (lines 121 -133)

- previous measures of disgust avoidance (lines 202 – 212). 

3) It is not clear to me, why the authors give so much attention to CFA of the model of four overlapping factors, if they correctly state that CFA is not an appropriate method for analysing inter-correlated factors. I understand the argument of comparability with other studies, but then the whole section on the CFA could be extensively shortened. 

In the analysis section of Study 2, we explain why we decided not to rely on CFA results (lines 537 – 543), which, as the reviewers also mention, we consider informative and critical for readers to comprehend why we focus mainly on the ESEM results. 

Finally, in the results section of Study 2, we only briefly report on the results of the CFA model (see lines 574 – 578). In order to make it more apparent that afterwards we only discuss the ESEM results, we separated the CFA and ESEM results into two paragraphs. 

We also made our emphasis on ESEM clearer in lines 420 - 423 & lines 534 - 535: 

“In Study 2 we therefore examined the factor structure of this overlapping 4-factor model, mainly by using exploratory structural equation modelling (ESEM; [29]), and its relationship with other constructs.”

“In the following, we examined the overlapping-factor model. In addition to CFA, we used Exploratory Structural Equation Modeling (ESEM; [29]), […]”.

Apart from that, we report the CFA-based factor loadings in Table 5, because we did not want to leave out this information for the reasons named in the analysis section (i.e., comparability with other studies). 

4) Given the nature of data collection (online study), the sample size seems a bit low. The authors should justify the sample size by conducting a power-analysis prior to the study. Also, they should report effect sizes for all their results. 

For both study 1 and study 2, we based our sample size calculations on general rules of thumb, as is common with factor analyses. We added statements explain the rationale for the sample sizes of Study 1 (lines 228 – 229) and Study 2 (lines 423 - 424).

“We aimed for a sample size of at least 400 participants, based on Fabrigar and colleagues [43] categorizing sample sizes of N > 400 as large.” 

“We aimed for a sample size of at least 500 participants, based on Comrey & Lee [55] categorizing a sample size of 500 as ‘very good’.”

Due to the lack of consensus about sufficient sample sizes for EFA, CFA and/or ESEM, and sample size calculations being dependent on the complexity of the model (e.g., number of factors, variables to factor ratio), it is difficult to conduct an a-priori power analysis (particularly when it is not known how many items are retained as it was the case in Study 1). Given that our models are not that complex, we considered using a general rule of thumb to be sufficient. 

Nonetheless, we mentioned our reliance on general rules of thumb as a possible limitation and recommend future research to conduct a-priori power analyses (as the variable to factor ratio is now known for the DAQ factor model; lines 788 - 791): 

“Lastly, the sample sizes of our studies were based on general rules of thumb instead of a-priori power analyses. Although we consider the general rules of thumb to be sufficient due to the relatively low complexity of our models, we recommend future research on the DAQ to conduct a-priori power analyses. “ 

With regard to effect sizes, we are not aware of a standard effect size measure of EFA/CFA/ESEM. We calculated and added the 90% confidence intervals around all reported RMSEA values, which provide an indication of effect size. 

5) I did not understand why the authors ran EFA and CFA separately for each factor and then for the four-factor model. I am not sure if this is a correct use of the method. Usually, factor analyses are run for the whole model (in this case with four factors), not separately for individual factors. 

For step 1 (item reduction), we ran (a) EFAs separately per factor (subscale) as a procedural step for reducing the number of items per factor, and (b) ran a 4-factor EFA on the combined item set (whole model) to examine the fit of the 4-factor factor structure on the reduced item set. No CFAs were involved here. We followed this procedure according to the advice of Prof. Dr. Timmerman (see acknowledgments), a statistician from our faculty and Professor in multivariate analyses. 

In the second step, we ran a CFA and an ESEM for the whole model (not per factor). 

In order to make our analysis/results clearer, we split Table 2 into two tables. Table 2 now presents the 1-factor EFA’s per subscale item set, and gives both the initial and reduced item numbering and the deleted items. Table 3 contains the 4-factor EFA that was run on the combined/overall items.

We also adapted the structure of the manuscript (separating study 1 & study 2; describing the analysis under ‘methods’; re-structuring the analysis & results section) and adapted the wording of the analysis section to help clarify the methodological details of the manuscript.

6) Throughout the whole manuscript, I did not find any results concerning mean sum scores of the DAQ, only factor scores are being reported. I think this should be included in the manuscript. The authors should also analyse and report how the main individual characteristics (gender, age, education) affect the DAQ scores. Unfortunately, this is also missing. 

We thank the reviewer for bringing this point to our attention. Based on the reviewer’s comment, we added the means of unweighted sum scores of the DAQ subscales overall and across different demographic variables, including gender, nationality, and field of study to the manuscript in the form of Table 7. We also reported the correlations of DAQ subscales with age.

We added a description of the analyses to the analysis section of Study 2 and reported the results in the results section of Study 2:

Analysis section (lines 569 - 571):

“Lastly, we calculated the means of unweighted sum scores of the DAQ subscales overall and across different demographic variables, including gender, nationality, and field of study. We also reported correlations between DAQ subscales scores and age.”

Results section (lines 629 - 632):

“Means of DAQ Unweighted Sum Scores 

The means of unweighted sum scores of the DAQ subscales overall and across different demographic variables (gender, nationality, & field of study) and correlations of unweighted sum scores of the DAQ subscales with age can be found in Table 7.”

1) Line 35-36: The sentence is grammatically incorrect and does not make sense, please revise.

In lines 34-36, we wrote “In contrast, the results related to the escape factor question the suitability of self-report to assess disgust escape.” In order to emphasize that the word “question” is acting as a verb in this sentence, we reformulated the sentence to “In contrast, the results related to the escape factor may call the suitability of self-report to assess disgust escape into question.”

2) Line 50-53: The list of disgust elicitors is much longer and the authors should provide more examples, e.g. small animals, political orientation,… and describe also magical thinking in relation to disgust.

We adapted the description of disgust elicitors to add more examples (lines 50 – 55). We also discussed the universality vs. context dependency of disgust elicitors (lines 55 - 57). In the context of disgust prevention, we referenced the laws of sympathetic magic (155 – 158).

3) Line 64-67: This sentence is just repeating what has been already said above (line 56-60). Also, please change “Research so far has identified,…” to “Research has so far identified…”.

We deleted the superfluous sentence and adapted the formulation in the previous sentence (line 68). 

4) Line 113-114: This sentence is a redundant as it is again a repetition of the previous one (e.g. line 85-86).

We deleted the sentence containing the repetition. 

5) Line 114-117: It is not correct to say that “pathogen disgust evolved to protect humans from disease-inflicting stimuli that cannot be seen or otherwise detected…”. If this had been the case, disgust could have been triggered by any stimulus. In fact, we can guess fairly well on the presence of pathogens, even if we cannot see them (rotten food, worms, insect, sick people, bodily fluids,…).

We thank the reviewer for sharing their observation with us. It is indeed correct that common pathogen-containing stimuli can be guessed (rotten food etc.). Pathogens themselves are not detectable, and can spread in various (unnoticed) ways (e.g., surfaces, air, insect bites). Therefore, innocuous stimuli (e.g., surfaces) can, in principle, contain pathogens and thus represent disease-inflicting stimuli. In order to make our argument clearer we adapted the sentence to (lines 150 - 152): “The perspective that pathogen disgust evolved to protect humans from pathogens that cannot directly be seen may partially explain why disgust is geared towards a better safe than sorry heuristic.” 

6) Line 136-137: Again, this sentence is not totally correct. Contamination is only one threat disgust responds to (next to disease transmission, intoxication,…).

We adapted the sentence to avoid specifying “threat of contamination” (lines 175 - 176): “Disgust escape is adaptive when it promotes people to distance themselves from situations in which a threat to the organism is imminent.”

7) Line 152: should be ‘help clarify’, please revise

We revised the wording according to the reviewer’s suggestion (line 189).

8) Line 160: please delete “promotes”

We deleted the word “promotes”. 

9) Line 175: Unless it is a requirement of PLoS One, I find the structure here very unusual. Why is ‘Method’ comprised of ‘Participants’ and separated from ‘Materials’? Generally, you should have a chapter ‘Material and Methods’ where ‘Participants’ should be the first subchapter followed by ‘Assessment’, ‘Procedure’, ‘Statistical Analysis’,… 

Visually it appears like ‘Materials’ is a new heading, but ‘Materials’ (font size 16) is actually a sub-heading of ‘Method’ (font size 18). However, we re-considered the set-up of the method and results. 

Initially, we attempted to combine Study 1 and Study 2 into one method section (due to overlap in sample recruitment methods and procedures), but realized that it may have led to a confusing structure of the paper (e.g., we included the analysis section in each step separately instead of in the end of the method section). We attempted to improve the comprehensibility of the manuscript by splitting it into Study 1 and Study 2. This also meant that instead of a joint method section, we added separate method sections for each study (which include the sections ‘participants’, ‘materials’, ‘procedure’, and ‘analysis’).

10) It is not clear to me who was included in Sample 2. 

We recruited participants of Study/Sample 2 from the same participant pools which we also used to recruit participant for Study/Sample 1. To make sure that we did not test the same participants, participants from Study 1 were not allowed to participate in Study 2. We adapted the wording of the sample description of Study 2 to make this clearer (lines 462 – 466):

“Like in Study 1, we recruited our sample via two university-based participant pools consisting of (Pool 1) first-year bachelor psychology students (n = 320; participation in exchange for course credit) and (Pool 2) a broader group of young adults (n = 193; participation in exchange for financial compensation: 4€) Participants of Study 1 were excluded from participating in Study 2.”

11) Line 212: please add ‘to’ after ‘(1)’

We added ‘to’ to the formulation (line 487). 

12) Line 247-248: This is the first time the authors mention ‘a subset of a domain-specific version of the DAQ’, but the reader has no idea, what it is. It should have been explained earlier with the description of the psychometrics used in the study. 

We thank the reviewer for pointing this out. We added a short description of the domain-specific version of the DAQ to the materials sections of Study 1 (lines 272 - 276): 

“The materials also included the initial item set (25) of the Body-related Disgust Avoidance Questionnaire (B-DAQ) which aims to assess people’s tendencies to avoid experiencing body-related disgust. The B-DAQ is a body-related version of the Disgust Avoidance Questionnaire (DAQ). The B-DAQ and related materials can be found on the OSF (https://osf.io/4mzfs/) and will not be described here because it would extend the scope of the paper.”

and Study 2 (lines 510 - 515): 

“The materials also included the reduced item set (18) of the Body-related Disgust Avoidance Questionnaire (B-DAQ) which aims to assess people’s tendencies to avoid experiencing body-related disgust, and questionnaires related to the B-DAQ. The B-DAQ is a body-related version of the Disgust Avoidance Questionnaire (DAQ). The B-DAQ and related materials can be found on the OSF (https://osf.io/4mzfs/) and will not be described here because it would extend the scope of the paper.”

13) Line 296, Analysis Plan: Why did the authors 13) Line 303: I am not sure what the authors mean by ‘unidimensional factors’. I think factors should always be unidimensional.

We replaced this term with ‘unidimensional factor models’ (lines 298 - 299): 

“The goal was to create unidimensional factor models, through the stepwise exclusion of ‘suboptimal’ items.” 

14) Line 494: Wouldn’t a Spearman correlation be better given the fact that the distribution of DAQ scores should deviate from normality as they are based on a Likert-scale items (as the authors correctly acknowledged earlier)? I also did not understand if the authors correlated sum scores or factor scores of the DAQ and other measures, it is not very clear from the text. 

We decided for Pearson’s correlations because variables showed no marked deviations from a normal distribution (see lines 603 - 604). We examined the results when using Spearman’s correlation coefficients, which were similar compared to results obtained using Pearson’s correlation coefficients (see Table R1 in this document for Spearman’s correlation coefficients). 

In the section on the relationship between the DAQ subscales and related constructs, we reported on the correlations of both unweighted (UW) sum scores (calculated by summing the respective items of each factor) and ESEM-weighted sum scores obtained using the ‘save=fscores’ function in Mplus) of the DAQ subscales and other measures. In the analysis plan of Study 2, we made it clearer that both sum score methods were examined in the correlational analyses (see lines 564 – 565): 

“We thus reported the correlations of both UW- and ESEM- sum scores of each subscale with the constructs listed above.”

Reviewer 2 

1) The goal of the present paper was to develop and initially validate a self-report measure of disgust avoidance. Overall, the paper is well-written and the idea behind the disgust avoidance construct is intriguing. The paper would benefit from a stronger justification for the development of the scale and need for this measure in the assessment and treatment of psychological disorders. There are also a number of analytic ambiguities that make it difficult to assess the quality of this measure. I’m not convinced of the factor structure and final measure. Below, I outline my concerns in more detail.

We would also like to thank the second reviewer for taking the time and effort to evaluate our manuscript. We appreciate the reviewer’s comments and we hope we were able to address the reviewer’s concerns adequately. Please note that line references to the changes made to the manuscript refer to the clean manuscript (in which all track changes were accepted).

2) The authors appropriately raise the similarity between experiential avoidance and disgust avoidance, particularly with regard to psychopathology. Is disgust avoidance just a specific form of experiential avoidance? As a relatively large body of research has linked experiential avoidance with various psychological disorders (all those mentioned in connection with disgust sensitivity), why is a disgust avoidance measure necessary? Why isn’t experiential avoidance sufficient? Although disgust has been related to a number of psychological disorders, I don’t think that it is fair to say that disgust avoidance is the only or primary form of avoidance underlying these disorders. It would be useful to distinguish these constructs further and provide justification for the new measure.

Disgust avoidance is proposed to be a specific form of experiential avoidance (i.e., specific to the emotion of disgust). We do not think or mean to say that disgust avoidance is the only form of avoidance in these disorders; it may be of primary importance in some disorders, but this remains to be tested in future research. By specifically measuring the avoidance of disgust, we might be able to gain insights into individual differences in disgust-based psychopathology, that could not be examined using a measure of general experiential avoidance (which assesses avoidance of negative affect in general, but not the avoidance of specific emotions). We addressed these points in lines 121 - 133. 

In addition, we emphasized the potential relevance of disgust avoidance throughout the introduction (also in accordance with comments provided by Reviewer 1), for example in lines 73 – 78 & 92 – 105 (pertaining to the relationships between disgust propensity/sensitivity and disgust avoidance), lines 106 – 120 (pertaining to the persistence of disgust associations), and lines 202 – 212 (pertaining to previous measures of disgust avoidance). 

Due to the number and the extent of changes to the introduction, we decided not to copy-paste the changes into the response letter, but to merely refer to lines in the manuscript. 

3) How exactly was the “broader sample of young adults” recruited? 

The broader sample of young adults was also recruited via one of the university-based participant pools. We re-structured the sentence and changed the wording to make this clearer in the sample description of study 1 (lines 232 - 235) and of study 2 (lines 462 – 465):

“[…], we recruited our sample via two university-based participant pools consisting of (Pool 1) first-year bachelor psychology students (n = 162[320]; participation in exchange for course credit) and (Pool 2) a broader group of young adults (n = 255[193]; participation in exchange for financial compensation: 2€/4€).”

Pool 2 is a university-based research participation pool which is open for anyone to register; nevertheless, it consists mainly of young adults who are currently studying or have been studying at the University in the past.

For both sample/study 1 and sample/study 2, we reported sample characteristics per participant pool (in addition to the overall sample), which can be found in Table 1 and Table 4.

4) A large proportion of Sample 2 (roughly a third, 251 out of 764) was excluded. How many fell into each reason for exclusion? What might explain the higher exclusion rate for Sample 2 compared to Sample 1? Did excluded participants differ from included participants with regard to demographics or any primary study variables? 

We thank the reviewer for pointing this out. Based on the reviewer’s suggestions, we added the numbers and percentages of participants being excluded from Sample 1 and Sample 2 per exclusion criterion. We speculate that the reason for the higher exclusion rate in Sample 2 was due to asking participants about their motivation to engage in the study properly in Study 2 but not Study 1. Since Study 2 was longer than Study 1 (30-45 minutes compared to 15 minutes), it is likely that a number of participants did not stay fully motivated when filling out the online survey. The added exclusion number can be found in lines 237 - 240 (Study 1) and lines 468 - 475 (Study 2):

 “From the initial n = 495, n = 78 (15.76%) participants were excluded, because they (a) did not consent to participate in the study/wanted to withdraw their responses from the study (n = 20; 25.64%), did not answer both control questions correctly (n = 58; 74.36%).”

“From the initial n = 764 participants, n = 251 (32.85%) participants were excluded, because they (a) did not consent to participating/to allowing the use of their data (n = 76; 30.28%), (b) did not answer both control questions correctly (n = 130; 51.79%), or (c) indicated that they were not motivated enough to properly engage in the study (n = 45; 17.93%). The percentage of excluded participants was higher in Study 2 than in Study 1 (15.76%), which might have been due to Study 2 excluding participants based on their motivation, which was not done in Study 1 (i.e., this question was not asked in Study 1).”

We decided not to conduct further analyses comparing excluded from included participants, because a number of excluded participants indicated that they did not consent to participating/allow the use of their data/wanted to withdraw their responses from the study, which does not allow us to use their data in any analyses. Nonetheless, we acknowledge that a substantial number of participants had to be excluded and that a selection bias cannot be ruled out. We also point to the importance of including control questions (as was done in the project) to improve the quality of the data and to limit noise related to random answers of non-motivated participants. We reflected this in lines 738 – 787 in the manuscript.

“We also would like to acknowledge that a substantial number of participants had to be excluded from our studies (particularly from Study 2) and that a selection bias cannot be ruled out. In the meantime, this also points to the importance of including control questions (as was done in the current studies) to improve the quality of the data and to limit noise related to random answers of non-motivated participants.”

5) How was sample size determined? 

For both study 1 and study 2, we based our sample size calculations on general rules of thumb, as is common with factor analyses. We added statements explain the rationale for the sample sizes of Study 1 (lines 228 – 229) and Study 2 (lines 423 - 424).

“We aimed for a sample size of at least 400 participants, based on Fabrigar and colleagues [43] categorizing sample sizes of N > 400 as large.” 

“We aimed for a sample size of at least 500 participants, based on Comrey & Lee [55] categorizing a sample size of 500 as ‘very good’.”

Due to the lack of consensus about sufficient sample sizes for EFA, CFA and/or ESEM, and sample size calculations being dependent on the complexity of the model (e.g., number of factors, variables to factor ratio), it is difficult to conduct an a-priori power analysis (particularly when it is not known how many items are retained as it was the case in Study 1). Given that our models are not that complex, we considered using a general rule of thumb to be sufficient. 

Nonetheless, we mentioned our reliance on general rules of thumb as a possible limitation and recommend future research to conduct a-priori power analyses (as the variable to factor ratio is now known for the DAQ factor model; lines 788 - 791): 

“Lastly, the sample sizes of our studies were based on general rules of thumb instead of a-priori power analyses. Although we consider the general rules of thumb to be sufficient due to the relatively low complexity of our models, we recommend future research on the DAQ to conduct a-priori power analyses. “ 

6) Given the claims in the intro that a measure of disgust avoidance would be useful for clinical purposes, it is strange that no measures of psychopathology were included. What was the reasoning behind the validity measures that were chosen and the exclusion of mental health measures? 

We agree that it may come as a surprise why no clinical measures were included. We added a paragraph explaining in more detail why the particular validity measures were chosen (lines 426 – 434):

“We aimed to examine relationships between the DAQ and other instruments aimed at measuring related constructs to evaluate the DAQ’s convergent validity. We chose to examine the association between DAQ subscale scores and other disgust-related individual difference measures (disgust propensity & sensitivity) as well as broader emotion-related scales (experiential avoidance & emotion regulation). As we argued earlier, we assume that trait disgust variables, experiential avoidance, and emotion regulation are conceptually related to the construct of disgust avoidance. Although we emphasized the potential clinical relevance of the DAQ, we did not to include clinical measures yet, because we decided to first focus on the DAQ’s construct validity before examining its criterion validity.”

We refer to the need for future research to establish the criterion validity of the DAQ, specifically related to clinical measures/outcomes, throughout the discussion (e.g., lines 691 - 692; lines 722 – 724; lines 741 - 743; lines 778 – 780) and in the abstract (lines 37 – 39).

7) What was the rationale for the questionnaire order in Sample 2? Any concerns about order effects? 

We added an explanation for the order of the questionnaires (lines 525 – 527): 

“We decided on this order because we wanted to present the questionnaires in blocks of similar themes (trait disgust scales, body-related scales, emotion regulation scales).” 

We considered the presentation of questionnaires in blocks of thematic similarity to be helpful in reducing potential confusion/ambiguity, thereby improving quality of the answers. We see no obvious reasons for undesirable order/carry over effects.

8) Were there any differences between participants based on recruitment method? 

We thank the reviewer for this question. In order to provide more clarification, we reported sample characteristics per recruitment pool (in addition to the overall sample), which can be found in Table 1 (Study 1) and Table 4 (Study 2). The sample characteristics appear to be reflective of differences between sample populations of the two recruitment pools (e.g., Pool 1 contains psychology students, a program in which the percentage of Germans is quite high). 

9) For Sample 1, were separate EFAs conducted with the a priori subscale items or was EFA conducted with all items? Based on Table 2, it seems like an EFA was conducted with all of the items, comparing a single and a four factor solution. Why not the proposed two-factor solution? However, the write-up makes it sound like separate EFAs were conducted with each set of subscale items. There needs to be clarification as to what analyses were exactly conducted.

In sample 1, we conducted separate EFAs on each of the (four) subscale item sets (for the item reduction). After item reduction, we ran a 4-factor EFA on the combined/overall items. 

We see how Table 2 added confusion, and decided to split it into two tables. Table 2 now presents the 1-factor EFA’s per subscale item set, and gives both the initial and reduced item numbering and the deleted items. Table 3 contains the 4-factor EFA that was run on the combined/overall items.

We also adapted the structure of the manuscript (separating study 1 & study 2; describing the analysis under ‘Methods’; re-structuring the analysis & results section) and adapted the wording of the analysis section to help clarify the methodological details of the manuscript.

10) p. 15 - It is very confusing when item numbers are referred to in-text, but those numbers do match item numbers in tables (e.g., Table 2). Keep item numbers consistent or provide item wording. Also, based on Table 2, it is unclear why certain items were retained when they seem to cross-load on factors or load weakly (e.g., item 4). Same issue with Table 3. 

We see how the item numbering was confusing. We added statements to make it clear which item set the item numbers refer to: 

“Please note that item numbers given here refer to the initial item set as presented in Fig 1a.” (lines 321 – 322). 

“Please note that item numbers referred in the description of the item reduction correspond to the initial 25-item set (see Table 2).” (lines 353 - 354)

We also adapted Table 2 by including the initial and reduced item numbers as well as the (wording of the) deleted items. In addition, we adapted Fig 1 to illustrate changes in factor structure and corresponding item numbers.

We added an explanation for why no further items were excluded based on the 4-factor EFA model (depicted in Table 3) in Study 1 (lines 385 - 386):

“We did not exclude any additional items at this stage because we did not set the factor loadings in the 4-factor EFA model as a criterion to exclude items.”

In the description of the results of study 2 (as depicted in Table 4), we already stated that “Because of the RMSEA > .08, the low target loadings in the ESC (and PREV) factor, and the negative correlation between ESC and BEH in the ESEM model, we explored possible adjustments to the model through item exclusions or allowing item errors to correlate. Because the different methods we explored did not yield adjusted models that performed better with regards to the issues named above (RMSEA’s > .08/low target loadings remained/negative correlations increased), we decided not to include any adjustments to the model structure.” (lines 588 -591).

11) p.18 – the shift in labels for the 4 factors is jarring and unclear. More explanation and justification for this change needs to be provided. 

We see how the description of the change in labels was unclear and adapted the text (lines 402 - 424):

“There are four concepts that we hypothesized to be underlying our statistical model: disgust prevention (PREV), disgust escape (ESC), behavioral disgust avoidance (BEH), and cognitive disgust avoidance (COG). These four concepts (PREV-ESC-BEH-COG) could be argued to represent two dimensions of disgust avoidance, namely focus (PREV vs. ESC) and strategy (BEH vs. COG). These two dimensions are assumed to be overlapping. In other words, in any case of disgust avoidance, both the dimension of focus (in the form of either prevention or escape) and of strategy (either behaviorally or cognitively) are assumed to be present. For example, avoiding to go into a situation which could elicit disgust represents both a focus (here: prevention) and a strategy (here: behavior). 

The problems of the 4-factor model we observed in the 4-factor EFA (study 1) and 4-factor CFA (study 2; see description above) might have arisen because the subscales of study 1 measured the overlapping concepts of PREV, ESC, BEH and COG. More specifically, Behavioral Prevention (BP) taps into the constructs of BEH and PRE, Cognitive Prevention (CP) assesses COG and PRE, Behavioral Escape (BE) assesses BEH and ESC, and Cognitive Escape (CE) assesses COG and ESC (see Fig 1c). Re-examining the scale as a whole, we would expect each item of the DAQ to fall on both dimensions of disgust avoidance and thus load on one type of focus (either PREV or ESC) as well as on one type of strategy (either BEH or COG). Based on this, the resulting model (see fig 1c) would form a 4-factor structure with overlapping factors: PRE (items 1-9), ESC (items 10-17), BEH (items 1-5 + 10-13), and COG (items 6-9 + 14-17). In study 2 we therefore examined the factor structure of this overlapping 4-factor model and its relationship with other constructs.”

In addition, we adjusted Fig 1 to make the changes in model structures (in both Study 1 and Study 2) clearer.

---

## [Decision Letter · Decision Letter 1]

9 Dec 2020

PONE-D-20-16240R1

Individual Differences in Avoiding Feelings of Disgust: Development and Construct Validity of the Disgust Avoidance Questionnaire

PLOS ONE

Dear Dr. von Spreckelsen,

Thank you for submitting your manuscript to PLOS ONE. After careful consideration, we feel that it has merit but does not fully meet PLOS ONE’s publication criteria as it currently stands. Therefore, we invite you to submit a revised version of the manuscript that addresses the points raised during the review process.

**Although both Reviewers have appreciated the Authors’ responsiveness to their review and their efforts to revise the manuscript, one of them still found that the statistical analysis has not been performed appropriately. In particular, references and explanations for the measurement development approach should be provided.**

We look forward to receiving your revised manuscript.

Kind regards,

Stefano Federici, Ph.D.

Academic Editor

PLOS ONE

Additional Editor Comments (if provided):

Although both Reviewers have appreciated the Authors’ responsiveness to their review and their efforts to revise the manuscript, one of them still found that the statistical analysis has not been performed appropriately. In particular, references and explanations for the measurement development approach should be provided.

Reviewers' comments:

Reviewer's Responses to Questions

**Comments to the Author**

1. If the authors have adequately addressed your comments raised in a previous round of review and you feel that this manuscript is now acceptable for publication, you may indicate that here to bypass the “Comments to the Author” section, enter your conflict of interest statement in the “Confidential to Editor” section, and submit your "Accept" recommendation.

Reviewer #1: All comments have been addressed

Reviewer #2: (No Response)

2. Is the manuscript technically sound, and do the data support the conclusions?

Reviewer #1: Yes

Reviewer #2: Partly

3. Has the statistical analysis been performed appropriately and rigorously? 

Reviewer #1: Yes

Reviewer #2: No

4. Have the authors made all data underlying the findings in their manuscript fully available?

Reviewer #1: Yes

Reviewer #2: Yes

5. Is the manuscript presented in an intelligible fashion and written in standard English?

Reviewer #1: Yes

Reviewer #2: Yes

6. Review Comments to the Author

Reviewer #1: I would like to thank the authors for taking such a great effort when revising the manuscript. They have meticulously and appropriately addressedd each of my comments. Thanks to their work, the manuscript has been much improved and I'm happy with it's current state. I have no further concerns.

Reviewer #2: Although I appreciate the authors’ responsiveness to my initial review and their efforts to revise the manuscript, I am still not convinced that this disgust avoidance measure is needed. As the authors have argued that this measure may be especially important for understanding psychopathology, evidence of this is necessary. In particular, it seems crucial to demonstrate that the disgust avoidance measure has predictive validity unique to or above and beyond experiential avoidance measures.

The stepwise approach to item reduction is unusual. Standard practice would be to conduct a series of EFAs with all of the items to identify factor structure and reduce items. I can’t say that I’ve seen a measurement development paper in which separate EFAs were conducted with each set of factor items. References for this approach should be provided and an explanation for why this approach, rather than standard approach, was taken. I still question the four-factor solution with items that cross-load. I would be curious to know if the factor structure and final set of items differ if a standard approach was utilized.

Given the homogeneity of the two samples, I question the generalizability of these findings. Again, I doubt the utility of this measure. Additional data with more diverse samples is required.

7. PLOS authors have the option to publish the peer review history of their article (what does this mean?). If published, this will include your full peer review and any attached files.

Reviewer #1: **Yes: **Jakub Polák

Reviewer #2: No

---

## [Author Response · Author response to Decision Letter 1]

22 Jan 2021

Dear Dr. Federici and Reviewer 2, 

On behalf of all authors, I hereby submit our further revised manuscript PONE- D-20-16240 "Individual Differences in Avoiding Feelings of Disgust: Development and Construct Validity of the Disgust Avoidance Questionnaire". We would like to thank you for evaluating our manuscript.

Below we describe in detail how we handled each of the remaining comments. Each comment is literally quoted, followed by our reply. Line references to the changes made to the manuscript refer to the clean manuscript (in which all track changes were accepted). 

We hope that you will consider the comments to be adequately addressed and find the revised manuscript suitable for publication in PLOS ONE.

Sincerely, also on behalf of the other authors,

Paula von Spreckelsen

Comment from the Editor:

Although both Reviewers have appreciated the Authors’ responsiveness to their review and their efforts to revise the manuscript, one of them still found that the statistical analysis has not been performed appropriately. In particular, references and explanations for the measurement development approach should be provided.

We are happy to see that both reviewers valued our revisions to the manuscript. With regard to the statistical analyses, we provided additional explanations and references to the method in our response to the reviewer below and in the manuscript. 

Before we turn to these changes, we would like to note that we adapted a small detail in the instruction of the DAQ (see S3 Appendix). In the instructions, we state: 

“This questionnaire will assess how people cope with situations or activities that can elicit disgust, for example: coming into contact with bodily fluids of another person, accidentally eating rotting food, seeing mutilated bodies on the TV, having sexual contact with someone you are not attracted to, hearing about incest, witnessing dehumanization or harm done to others.

For each of the statements presented below, please indicate the extent to which you agree or disagree with the statements.” 

We did not realize it at the time, but our example of a disgust elicitor “hearing about incest” may be considered offensive in some cultures. Because we do not wish to offend any culture, we thought it would be best to delete that example from the final questionnaire. We do not think that it would have an impact on our results, as we do not change any of the DAQ items, but merely omit one of the disgust-elicitors from the list of examples in the instructions. We adapted Supplementary material S3 by deleting the example and adding an explanation into the note of the Table.

Comments by Reviewer 2:

Although I appreciate the authors’ responsiveness to my initial review and their efforts to revise the manuscript, I am still not convinced that this disgust avoidance measure is needed. As the authors have argued that this measure may be especially important for understanding psychopathology, evidence of this is necessary. In particular, it seems crucial to demonstrate that the disgust avoidance measure has predictive validity unique to or above and beyond experiential avoidance measures.

We are happy that the reviewer appreciates our revisions. We firstly would like to say that we agree with the reviewer on the importance of examining the DAQ’s predictive validity. In our manuscript, we provide theoretical arguments about the DAQ’s potential relevance to the field of clinical psychology and voice that this potential relevance remains to be examined in future research (e.g., lines 37 – 39; lines 698 - 699; lines 729 – 731; lines 748 - 750; lines 779 – 787). Because we do not claim to have established the clinical relevance of the DAQ in the current project, we do not agree that it is crucial for the current paper to demonstrate said clinical relevance/predictive validity. The goal of the current manuscript is to introduce and to share our measure of disgust avoidance with the research community. We regard the project as a starting point for research into this subject with future research continuing to examine and evaluate its merit in psychopathology and possibly other fields of psychology (e.g., social psychology). It is common for questionnaires to be re-examined, optimized, and revised, with the goal to refine their validities and test their application and relevance to different domains. We therefore advocate that the judgement of the potential relevance or necessity of the DAQ is to be left to the research community.

The stepwise approach to item reduction is unusual. Standard practice would be to conduct a series of EFAs with all of the items to identify factor structure and reduce items. I can’t say that I’ve seen a measurement development paper in which separate EFAs were conducted with each set of factor items. References for this approach should be provided and an explanation for why this approach, rather than standard approach, was taken. I still question the four-factor solution with items that cross-load. I would be curious to know if the factor structure and final set of items differ if a standard approach was utilized.

We understand the reviewer’s concerns and would like to offer a more detailed explanation for the method which we chose to select the item pool of the DAQ in Study 1. It is indeed common to run an EFA on all items of a questionnaire when the goal is factor extraction. Our goals for the analyses in Study 1 were two-fold: Step 1) Item selection (EFAs on item sets; main goal); Step 2) ‘Factor extraction’ (EFA on all items; secondary goal). We decided on that order because our goal was to first create a proper item set before evaluating examining the factor structure (e.g., suboptimal/problematic items could have influenced the results of a factor extraction). The EFAs on the item-sets were therefore part of the item selection, which can be based both on statistical and/or ‘judgmental’ criteria (e.g., [1]), but were not part of a ‘factor extraction’. Item selection is part of any questionnaire development. We simply included it as a statistical step in our manuscript, which might have led to some confusion. 

As is done in item reduction (see [2] on their item reduction of facets of the JDI [Job Descriptive Index] using principal component analysis; see [3] describing the method of selecting items with the highest loading on a common factor underlying the items to guide item selection), we examined whether the items of one subscale loaded on one common factor. This step could have also been done in a CFA framework by using 1-factor CFA models. We decided to use EFA however because it allowed us to examine multi-factor models (next to 1-factor models) for a more exhaustive picture of item loadings. In other words, because our use of EFAs in this step was theory-driven, the distinction between EFA and CFA more or less vanished. After the item reduction, we then ran a 4-factor EFA to examine whether item loadings were in line with our hypothesized subscales. 

We added a short introduction to the statistical approach we used in Study 1 (lines 226 - 235):

“The main goal of Study 1 was to select items for the DAQ using both ‘judgmental’/evaluative (e.g., item content, wording, etc.) and statistical criteria (e.g., item loadings, reliability estimates; cf. [43]). We first compiled a list of potential items for the DAQ (based on judgmental criteria) and subsequently condensed it through a stepwise item reduction. The goal of the step-wise item reduction was to find a coherent item set per hypothesized subscale of the DAQ and it was based mainly on statistical criteria. More specifically, we used single- and multi-factor EFA (exploratory factor analysis; [44]) models and fitted them on the items of each hypothesized subscale to exclude ‘suboptimal’ items with the goal to create unidimensional factor models per subscale. As a last step, we fitted an EFA on all items to examine whether the item loadings were in line with our hypothesized subscales.”

As we believe that the method we chose was suitable for item reduction, we have confidence in the final item sets of the DAQ. With regard to the final factor structure, we don’t see reasons for why it would have been different using a different methodology, because we see strong theoretical arguments for the overlapping factor structure. That being said, every methodology comes with limitations. In our case, we strongly based item reduction on statistical criteria (i.e., examining whether items loaded well on one common factor), which may have made our results more sample-dependent [3]. In addition, our method may also have resulted in a narrow item content [3]. However, our overall item selection was based on both statistical and judgmental criteria (the latter was used to compile the initial list of items & helped in item reduction for difficult cases), which may have somewhat protected us from these limitations. Nonetheless, the extent to which the findings generalize to other samples remains to be examined (which we added to our discussion).

We reflected on the potential limitations of our item selection methodology in the discussion section (lines 790 – 793):

“Although we combined both judgmental and statistical criteria (e.g., in our item selection in Study 1) in order to decrease the sample-dependency of our results, it remains to be examined whether our results can be replicated in different samples.”

Given the homogeneity of the two samples, I question the generalizability of these findings. Again, I doubt the utility of this measure. Additional data with more diverse samples is required.

We agree that more diverse samples are required to establish the general applicability of the DAQ. In our discussion section, we make explicit that future research is needed to establish the DAQ’s general applicability due to our restricted/homogeneous samples (lines 788 – 790). As we do not claim to have established a general applicability of the DAQ, we do not agree that it is crucial for the current paper to include additional data. As we already stated above, we regard the current paper as a starting point for future research refining its validity and generalizability.

However, in order to make it more explicit that future research into the generalizability of the DAQ is needed, we added the following to the abstract (lines 36 – 37):

“In light of the exploratory nature of the project, future examinations of the DAQ’s validity and applicability to more diverse samples are essential.”

References

1. Wieland A, Durach CF, Kembro J, Treiblmaier H. Statistical and judgmental criteria for scale purification. Supply Chain Management. 2017;22(4):321-328. doi: 10.1108/SCM-07-2016-0230

2. Stanton JM, Sinar EF, Balzer WK, Smith PC. Issues and strategies for reducing the length of self report scales. Pers. Psychol. 2002;55(1):167-194. doi: 10.1111/j.1744-6570.2002.tb00108.x

3. Widaman KF, Little T, Preacher KJ, Sawalani GM. On creating and using short forms of scales in secondary research. In: Trzesniewski KH, Donnellan MB, & Lucas RE, editors. Secondary data analysis: An introduction for psychologists. American Psychological Association. 2011. p. 39-61 doi: 10.1037/12350-003

---

## [Editor Report · Decision Letter 2]

23 Feb 2021

Individual Differences in Avoiding Feelings of Disgust: Development and Construct Validity of the Disgust Avoidance Questionnaire

PONE-D-20-16240R2

Dear Dr. von Spreckelsen,

We’re pleased to inform you that your manuscript has been judged scientifically suitable for publication and will be formally accepted for publication once it meets all outstanding technical requirements.

Kind regards,

Stefano Federici, Ph.D.

Academic Editor

PLOS ONE
---

## [Editor Report · Acceptance letter]

1 Mar 2021

PONE-D-20-16240R2 

Individual Differences in Avoiding Feelings of Disgust: Development and Construct Validity of the Disgust Avoidance Questionnaire 

Dear Dr. von Spreckelsen:

I'm pleased to inform you that your manuscript has been deemed suitable for publication in PLOS ONE. Congratulations! Your manuscript is now with our production department. 

Kind regards, 

on behalf of

Prof. Stefano Federici 

Academic Editor

PLOS ONE